# Virally programmed extracellular vesicles sensitize cancer cells to oncolytic virus and small molecule therapy

Marie-Eve Wedge [1,2,16], Victoria A. Jennings[1,3,4,16], Mathieu J. F. Crupi[1,5,16], Joanna Poutou[1,5,16], Taylor Jamieson [1,5], Adrian Pelin[1,5], Giuseppe Pugliese [1], Christiano Tanese de Souza[1], Julia Petryk[1], Brian J. Laight [1], Meaghan Boileau[1], Zaid Taha [1,5], Nouf Alluqmani[1,5], Hayley E. McKay [1,5], Larissa Pikor[1], Sarwat Tahsin Khan[1], Taha Azad[1,5], Reza Rezaei[1,5], Bradley Austin[1], Xiaohong He[1], David Mansfield[3], Elaine Rose[1,5], Emily E. F. Brown [1], Natalie Crawford[1], Almohanad Alkayyal[1,6], Abera Surendran[1,5], Ragunath Singaravelu [1,5], Dominic G. Roy [1,5], Gemma Migneco[4], Benjamin McSweeney[1], Mary Lynn Cottee[1], Egon J. Jacobus[1,7], Brian A. Keller[1,5], Takafumi N. Yamaguchi[8], Paul C. Boutros [8,9,10,11], Michele Geoffrion[12], Katey J. Rayner[5,12], Avijit Chatterjee[13], Rebecca C. Auer[1,5,14], Jean-Simon Diallo [1,5], Derrick Gibbings [2], Benjamin R. tenOever [15], Alan Melcher[3], John C. Bell [1,5] & Carolina S. Ilkow [1,5✉]

Recent advances in cancer therapeutics clearly demonstrate the need for innovative multiplex therapies that attack the tumour on multiple fronts. Oncolytic or "cancer-killing" viruses (OVs) represent up-and-coming multi-mechanistic immunotherapeutic drugs for the treatment of cancer. In this study, we perform an in-vitro screen based on virus-encoded artificial microRNAs (amiRNAs) and find that a unique amiRNA, herein termed amiR-4, confers a replicative advantage to the VSVΔ51 OV platform. Target validation of amiR-4 reveals ARID1A, a protein involved in chromatin remodelling, as an important player in resistance to OV replication. Virus-directed targeting of ARID1A coupled with small-molecule inhibition of the methyltransferase EZH2 leads to the synthetic lethal killing of both infected and uninfected tumour cells. The bystander killing of uninfected cells is mediated by intercellular transfer of extracellular vesicles carrying amiR-4 cargo. Altogether, our findings establish that OVs can serve as replicating vehicles for amiRNA therapeutics with the potential for combination with small molecule and immune checkpoint inhibitor therapy.

[1] Centre for Innovative Cancer Therapeutics, Ottawa Hospital Research Institute, Ottawa, Ontario, Canada. [2] Department of Cellular and Molecular Medicine, University of Ottawa, Ottawa, Ontario, Canada. [3] Institute of Cancer Research, London, UK. [4] Leeds Institute of Medical Research at St James's, University of Leeds, Leeds, UK. [5] Department of Biochemistry, Microbiology and Immunology, University of Ottawa, Ottawa, Ontario, Canada. [6] Department of Medical Laboratory Technology, Faculty of Applied Medical Sciences, University of Tabuk, Tabuk, Saudi Arabia. [7] Department of Oncology, University of Oxford, Oxford, UK. [8] Jonsson Comprehensive Cancer Center, University of California, Los Angeles, Los Angeles, CA, USA. [9] Department of Urology, University of California, Los Angeles, Los Angeles, CA, USA. [10] Institute for Precision Health, University of California, Los Angeles, Los Angeles, CA, USA. [11] Department of Human Genetics, University of California, Los Angeles, Los Angeles, CA, USA. [12] University of Ottawa Heart Institute, Ottawa, Ontario, Canada. [13] The Ottawa Hospital, Division of Gastroenterology, Ottawa, Ontario, Canada. [14] Department of Surgery, University of Ottawa, Ottawa, Ontario, Canada. [15] Department of Microbiology, Icahn School of Medicine at Mount Sinai, New York, NY, USA. [16] These authors contributed equally: Marie-Eve Wedge, Victoria A. Jennings, Mathieu Crupi, Joanna Poutou. ✉email: cilkow@uottawa.ca

Cancers are ecosystems in which tumour cells communicate with their malignant counterparts and supporting cells either through direct contact or via the secretion of growth factors, metabolites and assorted extracellular vesicles (EVs) (e.g. exosomes and microvesicles)[1–4]. EVs are known to transmit information between normal cells and these pathways are usurped by malignant cells to promote their own growth, survival and resistance to therapeutic intervention[3,4]. Pathogenic viruses have also evolved strategies to use EVs as a means to transfer virally encoded gene products to neighbouring uninfected cells to enhance virus growth, spread and persistence in normal tissues[5–7]. We reasoned that it may be possible to design cancer lysing or oncolytic viruses (OVs) that exploit this EV information transmission network to improve OV therapeutic activity.

While there have been multiple strategies employed to restrict the replication of OVs to cancer cells, one common theme is the impaired antiviral response of most if not all tumours[8]. Normal cells have multilayered systems for detecting and responding to invasion by pathogens, many of these based upon networks that interface or overlap with pathways controlling cell growth, cell fate, metabolism and immune surveillance programmes[9–11]. As cancers evolve, they acquire genetic or epigenetic changes that allow them to overcome the growth restrictions that normal cells experience, and concurrently, lose some of the cell-autonomous systems responsible for detection and response to virus infections. The extent of antiviral programme loss in cancers is variable and OVs, like all other cancer therapeutics, will be more or less effective in a tumour depending upon the malignant cell's genetic and epigenetic makeup.

We previously demonstrated that selective OV killing of cancer cells can be enhanced by combined treatment with an epigenetic modifying drug[12]. These experiments led us to speculate that a synthetic sensitisation strategy could be exploited to create a gene expression profile in cancer cells that strongly favours OV replication. We hypothesised that antiviral systems in normal cells have sufficient redundancy to tolerate the reduced expression of specific antiviral gene products, but the decreased expression of these same genes in cancer cells would lead to enhanced OV growth and killing. To this end, we used a replication-competent library of viral recombinants individually enabled with the capacity to elicit RNA interference (RNAi) on host genes. Here, we show that the selection of improved replication in cancer cells helps to identify artificial microRNAs (amiRNAs) that could then be used to improve OV activity. We find that these "virus-sensitising" amiRNAs are transmitted from infected tumour cells to neighbouring uninfected cells, grooming them for an impending OV infection and enhanced cancer cell death.

## Results

### Selection of an artificial microRNA that enhances oncolytic virus anticancer potency.
Most RNA-based cytoplasmic viruses do not naturally encode microRNAs (miRNAs), yet can be engineered to express high levels of functional miRNAs[13]. As the specificity and potency of miRNA targeting is a product of its sequence, one can exploit this activity to "re-wire" the hairpins and perform an RNAi screen in the context of a bona fide infection to identify host restriction factors that impede replication in normal tissues[14,15]. Using a variation of this strategy, we screened for amiRNAs that selectively improved virus replication in cancer cells. A replicating Sindbis virus (SV) library encoding amiRNAs targeting some 16,000 mammalian genes was used to infect a panel of human and mouse cancer cell lines, cancer-associated fibroblasts (CAFs) and normal skin fibroblast cultures (Fig. 1a). Next-generation sequencing (NGS) analysis identified a selection of five amiRNA-encoding viral candidates that had a substantial fold enrichment in tumour cells following passaging,

compared to the input library (Fig. 1b and Supplementary Table 1). To test the ability of these amiRNAs to enhance OV replication and oncolysis, selected amiRNAs were cloned into the clinically staged oncolytic rhabdoviruses Vesicular stomatitis virus VSVΔ51[16] and Maraba virus[17] platforms (Supplementary Fig. 1a) and are herein arbitrarily named from 1 to 5. The sense amiRNA strand expression following infection of cancer cells was confirmed by RT-qPCR (Supplementary Fig. 1b). The ability of these amiRNA-expressing viruses to enhance OV replication was tested in a panel of cancer cell lines, CAFs and normal fibroblasts (GM38). Assessment of cell viability revealed that among all the selected amiRNAs, amiR-4 induced significantly greater cell death compared to control the virus in tumour cell lines (i.e., 786-0, BxPC-3, HPAF-II) and CAFs that were otherwise highly resistant to VSVΔ51-mediated killing (Fig. 1c and Supplementary Fig. 1c–e). Importantly, expression of amiR-4 did not lead to significantly enhanced killing by VSVΔ51 in healthy GM38 fibroblasts compared to virus control 72 h post infection (hpi) (Supplementary Fig. 1f). Furthermore, an increase in virus replication in vitro in the same cell lines used for the SV-amiRNA library screen (MIA PaCa-2, PANC1 and PANC02 pancreatic cancer cell lines) (Supplementary Fig. 1g) and ex vivo (pancreatic cancer patient-derived tumour cores) (Fig. 1d) cancer model systems was evident following VSVΔ51-amiR-4 infection compared to a control VSVΔ51 virus expressing a non-targeting control amiRNA (VSVΔ51-amiR-NTC). amiR-4 was found to significantly enhance virus-induced cytotoxicity in a comprehensive panel of human and mouse tumour cell lines but not in normal fibroblasts (Fig. 1e).

### amiR-4 targets cellular factors involved in epigenetic regulation and cytoskeleton stability.
Computational miRNA target prediction pipelines (TargetScanHuman and BLAST) were used to determine potential cellular gene products targeted by amiR-4 (Supplementary Tables 2 and 3). We tested the ability of VSVΔ51-amiR-4 to selectively reduce the expression of each candidate gene product at both the mRNA and protein levels. We found that the gene products encoded by ARID1A, PLEC and HDAC4 but not MCM2 were specifically decreased following infection with VSVΔ51-amiR-4 as shown by RT-qPCR (Fig. 2a) and immunoblotting analysis (Fig. 2b). Our results suggest that amiR-4 might potentiate OV-growth and cancer cell cytotoxicity via the downregulation of ARID1A, HDAC4 and PLEC.

### ARID1A plays a role in resistance to oncolytic virus infection.
The observation that decreased expression of the PLEC gene enhanced virus replication is consistent with earlier studies demonstrating that manipulation of cytoskeletal components enhances VSVΔ51 replication and oncolysis in various tumour models[18,19]. Similarly, multiple reports have previously demonstrated that the inhibition of histone deacetylases boosts the replication of various OV platforms[20–23]. On the other hand, the protein encoded by ARID1A is a subunit of the SWI/SNF chromatin remodelling complex which facilitates the access of transcription factors to DNA and is not previously known to have antiviral activity. Using CRISPR-Cas9 gene editing, the ARID1A was deleted from 786-0 and PANC1 cancer cells (Supplementary Fig. 2a, b). Cells lacking the ARID1A gene are significantly more sensitive than their parental counterparts to VSVΔ51 infection (P value < 0.0001 at all MOIs) (Fig. 2c and Supplementary Fig. 2c). As one would predict if the ARID1A mRNA is one of the primary targets of amiRNA-4, VSVΔ51 expressing this artificial microRNA did not substantially improve killing of ARID1A deleted cells (Fig. 2d and Supplementary Fig. 2d). ARID1A-knockout cells also display greater susceptibility to other OV platforms including

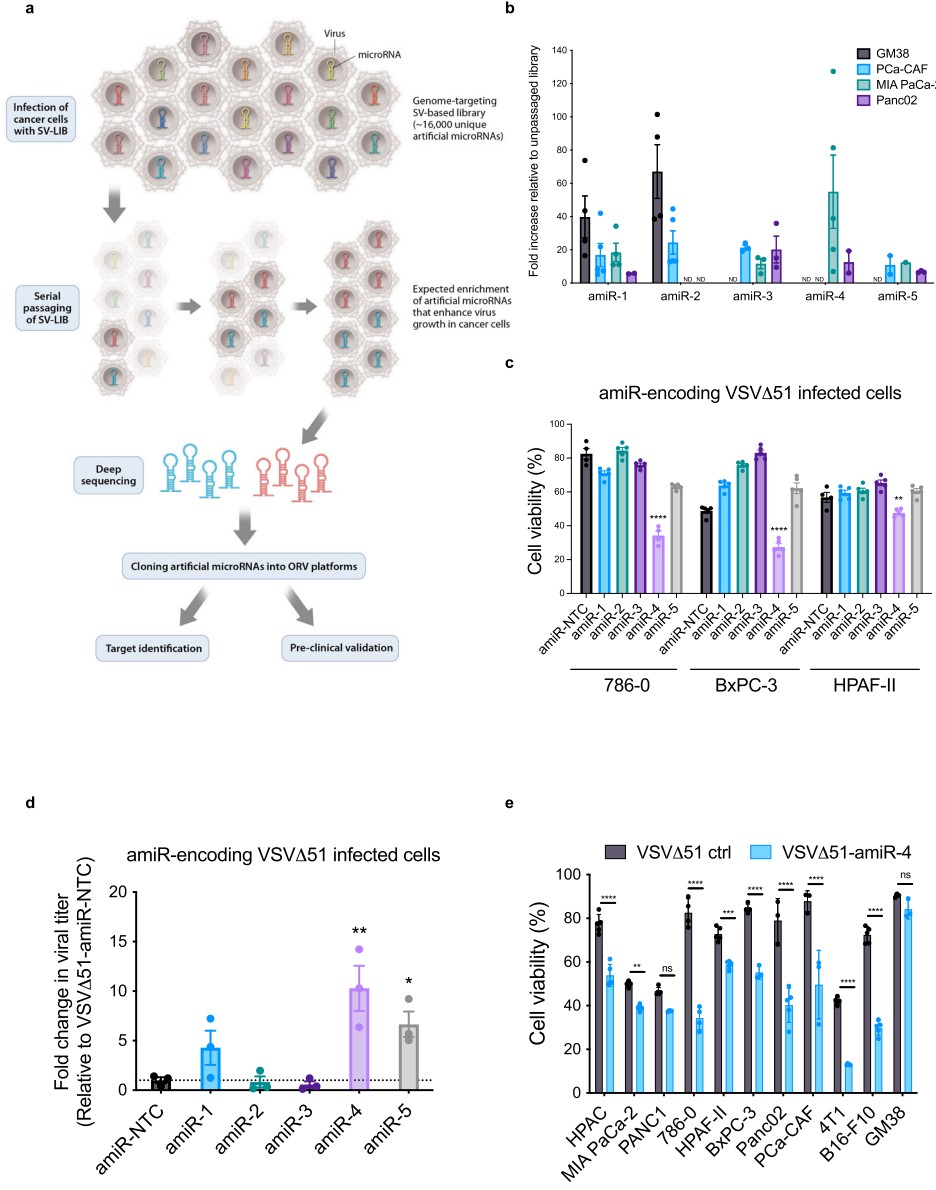

**Fig. 1 Screen of a Sindbis virus-based artificial microRNA library identifies artificial miRNA-4 as a pro-oncolytic virus amiRNA sequence. a** Schematic flowchart for the selection of pro-viral amiRNAs under pancreatic tumour cell selection. **b** Fold increase in deep sequencing amiRNA hits of the top five most enriched amiRNAs, compared to the unpassaged library in GM38, pancreatic (PCa) CAF-like fibroblasts, MIA PaCa-2 and Panc02 cells. Data displayed as mean ± SEM of five technical replicates. Full targeted RNA-sequencing datasets are included in Supplementary Data 1. **c** Relative percentage (%) of cell viability of 786-0, BxPC-3 and HPAF-II cells infected for 48 h (786-0 and BxPC-3) or 72 h (HPAF-II) with indicated amiRNA-expressing viruses at MOIs 1, 5 and 3, respectively, compared to uninfected cells. Results displayed as mean ± SEM of five biological replicates. Differences between amiRNA-expressing viruses and control virus were assessed by two-way ANOVA with Dunnett's multiple comparisons test (95% confidence intervals [CI]), **$P = 0.0048$, ****$P < 0.0001$. **d** Relative fold change in titers of amiRNA-expressing VSVΔ51 viruses compared to VSVΔ51-amiR-NTC following ex vivo infection of patient-derived pancreatic tumour cores. Data represent mean values ± SEM of three biological replicates. One-way ANOVA with Dunnett's multiple comparisons test (95% CI), *$P = 0.0392$, **$P = 0.0013$. **e** Cell viability of indicated cell lines after VSVΔ51-amiR-NTC (VSVΔ51 ctrl) or VSVΔ51-amiR-4 infection (MOI 0.1; 48 h) compared to mock-infected cells was measured using the alamarBlue® Assay. Data indicate the mean ± SEM of five biological replicates. Two-way ANOVA with Sidak's multiple comparisons test (95% CI), ns$P > 0.05$, **$P = 0.003841$, ***$P = 0.000125$, ****$P < 0.001$. Source data are provided as a Source data file.

a vaccinia virus (VV-TK−VGF−), an oncolytic Herpes Simplex Virus 1 (oHSV-1), SV, and Reovirus data (Fig. 2e–h and Supplementary Fig. 2e–g). Interestingly, we observed a replicative advantage of VSVΔ51-amiR-4 in *ARID1A*-expressing pancreatic cancer patient-derived samples using our unique biobank of pancreatic cancer patient-derived xenografts (Fig. 2i, j). These pancreatic tumours may have *ARID1A* gene alterations, similar to TCGA patient data showing the low frequency of gene mutations

or deep deletions (Supplementary Fig. 2h). Taken together, these results suggest that the ARID1A protein has a previously unappreciated role in mediating cellular antiviral programmes. To determine the changes in gene expression of *ARID1A*-knockout cells that could lead to virus sensitisation, we conducted RNA-sequencing (RNA-seq) analysis. Gene-Ontology (GO) term analysis revealed that some primary signalling programmes repressed in *ARID1A*-deficient cells are related to type I and II interferon

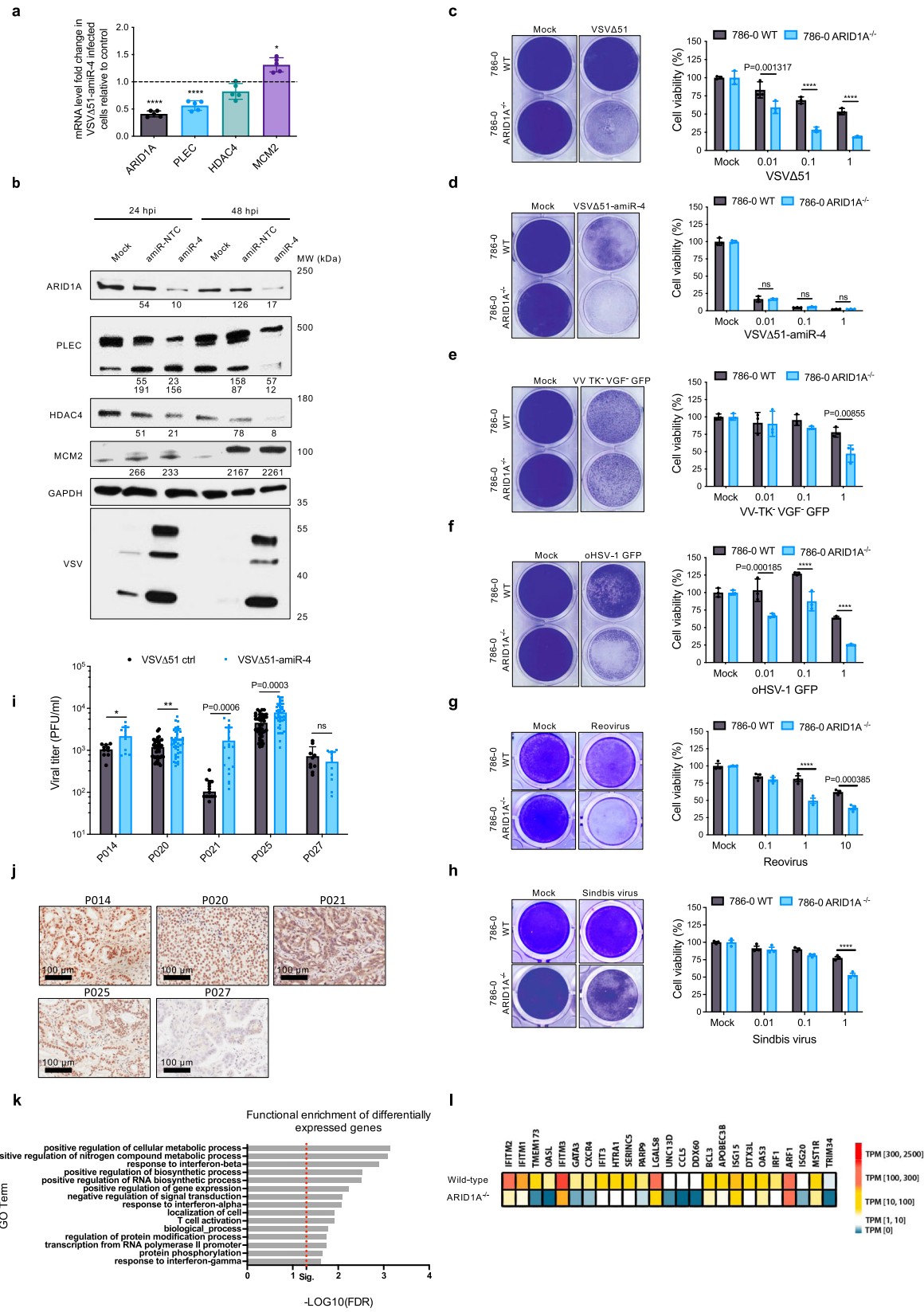

(IFN) signalling circuits (Fig. 2k, l and Supplementary Fig. 2i). A selection of these repressed genes was validated using RT-qPCR (Supplementary Fig. 2j). Together, these data indicate that an *ARID1A* wild-type status is associated with resistance to OV therapy and supports the use of an OV encoding amiR-4 to repress antiviral responses.

**Oncolytic virus-mediated targeting of ARID1A creates a synthetic lethal phenotype.** *ARID1A* represents a particularly interesting amiR-4 target since its deletion/mutation sensitises ovarian cancer cells to EZH2 methyltransferase inhibition with the small molecule GSK126 in a synthetic lethal fashion[24]. While synthetic lethality represents an attractive approach to the

**Fig. 2 Functional identification of ARID1A as a robust antiviral mediator. a** Fold change in expression of amiR-4 target genes determined by RT-qPCR in MIA PaCa-2 cells infected for 36 h with VSVΔ51-amiR-NTC or VSVΔ51-amiR-4 (MOI 1) compared to virus control and normalised to loading control (Rplp0). Two-tailed unpaired $t$ test, $*P = 0.0107$, $****P < 0.0001$. Shown are mean values ± SEM of five biological replicates. **b** Immunoblot analysis showing protein expression levels of amiR-4-predicted targets, GAPDH (loading control), and VSVΔ51 proteins in PANC1 cells, subjected or not to VSVΔ51-amiR-NTC or VSVΔ51-amiR-4 infection for 24 h and 48 h ($n = 3$). **c–h** Crystal violet staining pictures and cell viability measured by alamarBlue® Assay of 786-0 wild-type (WT) or ARID1A-knockout (ARID1A$^{-/-}$) cells mock-infected or infected at multiple MOIs for 48 h with multiple oncolytic virus platforms. Crystal violet staining pictures represent ($n = 3$) cells mock-infected or infected with VSVΔ51-GFP at MOI 1 (**c**), VSVΔ51-amiR-4 at MOI 1 (**d**), VV TK$^-$ VGF$^-$ at MOI 1 (**e**), oHSV-1 GFP at MOI 0.1 (**f**), Reovirus at MOI 1 (**g**), or SV-GFP at MOI 1 (**h**). Two-way ANOVA Tukey's multiple comparison test (95% CI), $^{ns}P > 0.05$, $****P < 0.0001$. Shown are mean values ± SEM of biological triplicates. **i** VSVΔ51-amiR-NTC or VSVΔ51-amiR-4 titers of ex vivo infected tumour cores obtained from several pancreatic cancer PDXs. Results displayed as mean ± SEM (P014, $n = 10$; P020, $n = 40$; P021, $n = 29$; P025, $n = 52$; P027, $n = 10$), two-tailed unpaired $t$ test of individual patient plots, $^{ns}P > 0.05$, $*P = 0.0213$, $**P = 0.007$. **j** Immunohistochemistry staining of ARID1A protein expression in five pancreatic cancer patient-derived xenografts (PDXs). Scale bar = 100 μm; ($n = 2$ technical replicates per PDX). **k** Gene-Ontology (GO) analysis of biological processes differentially expressed between PANC1 wild-type and ARID1A-knockout cells. Illustrated GO-terms represent all significantly different biological processes (Fisher's exact test) after correction for multiple hypothesis testing (FDR). **l** Heatmap showing gene transcript expression levels (Log2 TPM [transcripts per kilobase million]) of antiviral genes in uninfected PANC1 wild-type or ARID1A-knockout cells. Source data are provided as a Source data file.

selective targeting of cancer cells its actual application in clinical settings has proven to be limited. We hypothesised that an amiR-4-expressing OV could be combined with GSK126 to potentiate tumour killing within the tumour microenvironment. In fact, combining an amiR-4-expressing virus at very low multiplicities of infection (MOI) with GSK126 drastically increases cell death in several cancer cell lines and spheroid cultures (Fig. 3a–d and Supplementary Fig. 3a). As expected, the advantage of combining an amiR-4-expressing VSVΔ51 platform and GSK126 is lost in an ARID1A-deficient ovarian cancer cell line (SKOV3)[24] (Supplementary Fig. 3b). Together, our data suggest that the combination of an OV backbone armed with amiR-4 and the small-molecule inhibitor GSK126 facilitates tumour cell death via a synthetic lethal interaction between ARID1A and EZH2.

**Virally expressed amiR-4 is transmitted between cells by small extracellular vesicles.** While we observed enhanced killing with the OV-amiR-4 and GSK126 combinatorial treatment compared to either treatment alone (Fig. 3a, b), GSK126 treatment did not increase viral titers (Supplementary Fig. 3c–f) or the number of infected cancer cells as assessed by flow cytometry (Supplementary Fig. 3g–i). These data suggest that the impact of virally encoded amiR-4 extends beyond the initially infected cell. Small extracellular vesicles (SEVs) (e.g. exosomes) are membrane vesicles (~30–100 nm) secreted by most cells and have been shown to be natural carriers of functional proteins and genetic material, including miRNAs[25,26]. In agreement with previous studies[27,28], we found that SEV secretion by cancer cells is increased upon oncolytic rhabdovirus infection (Fig. 4a, b and Supplementary Fig. 4a). We hypothesised that the observed bystander killing of neighbouring uninfected cancer cells could be mediated by amiR-4-containing SEVs derived from VSVΔ51-amiR-4-infected cells. To test this hypothesis, we created an amiR-4-encoding rhabdovirus (MRBΔG) that lacks the viral G protein gene and can thus only spread when G protein is provided in trans. In this way, we can generate infected cells that express amiR-4 from the viral genome but cannot produce infectious virions. Thus, we were able to generate SEV preparations from infected cells that are devoid of "contaminating" infectious virions (Supplementary Fig. 4b–d). SEVs derived from MRBΔG-amiR-4-infected cells contain amiR-4 (Fig. 4c and Supplementary Fig. 4e, f) and these SEVs are taken up by uninfected cells (Fig. 4d). When naive uninfected cells were educated with amiR-4-containing SEVs, cell death was observed and cytotoxicity was enhanced in a synthetic lethal fashion with the combination of GSK126 (Fig. 4e, f and Supplementary Fig. 4g).

Rab27 GTPases mediate SEV release[29,30]. To further validate the transfer of virally encoded amiR-4 to uninfected cells through SEVs, we generated a Rab27a knockout 4T1 cell line and

demonstrated that these cells had greatly reduced SEV secretion (Supplementary Fig 4h, i), as has been previously shown with shRNAs directed against Rab27a[30,31], without significantly affecting viral replication ($P$ value > 0.5 at all MOIs) (Supplementary Fig. 4j). Mouse 4T1 cancer cells reprogrammed with SEVs passively transferred in transwell coculture system from Rab27a-depleted 4T1 cells infected with MRBΔG-amiR-4 (Fig. 4g) contained significantly less amiR-4 than cells exposed to the infected wild-type counterpart cells ($P$ value = 0.0066) due to reduced SEV production in the Rab27a-depleted 4T1 cells (Fig. 4h). Furthermore, the combinatorial treatment of VSVΔ51-amiR-4 and GSK126 resulted in a significant decrease in cancer cell viability ($P$ value = 0.0003), Rab27a-depletion in 4T1 cells abolished these effects ($P$ value = 0.7193) (Fig. 4i, j). Again, viral replication was unaffected by Rab27a deficiency or GSK126 treatment (Supplementary Fig. 4k, l). These data provide evidence that the transfer of virally expressed amiRNA-4 from infected to uninfected cells via SEVs contributes in part to the bystander sensitisation of uninfected cells to GSK126.

**amiR-4 enhances the oncolytic capacity of VSVΔ51 in in vivo tumour models.** To determine whether amiR-4 enhanced oncolytic activity in vivo, immunocompromised mice bearing human pancreatic HPAF-II subcutaneous (SC) tumours were treated with either VSVΔ51-amiR-NTC or VSVΔ51-amiR-4. A significant increase ($P$ value = 0.014) in viral output was observed after 48 h in tumours treated with VSVΔ51-amiR-4 compared to VSVΔ51-amiR-NTC control, indicating that the expression of amiR-4 within tumours enhances tumour-associated viral replication in vivo (Fig. 5a) without compromising either tumour specificity or viral biodistribution (Supplementary Fig. 5a). To determine whether the enhanced viral replication also occurs in the presence of an intact immune system, immune-competent mice bearing syngeneic TH04 pancreatic, or B16-F10 melanoma tumours were treated with VSVΔ51-amiR-4 or control for 24 h and 48 h. We similarly detected increased titers at both time points (Fig. 5b, c), suggesting that immune cells do not comprise the ability of VSVΔ51-amiR-4 to replicate better than VSVΔ51 control in tumours.

To assess whether the enhanced viral replication resulted in tumour debulking, animals bearing HPAF-II SC tumours were treated with vehicle control (PBS), VSVΔ51 control or VSVΔ51-amiR-4. VSVΔ51-amiR-4 enhanced tumour control compared to treatment control (Fig. 5d). Survival of immune-competent mice bearing syngeneic orthotopic TH04 pancreatic tumours upon VSVΔ51-amiR-4 treatment was also assessed (Fig. 5e). While this tumour model is typically extremely aggressive and relatively resistant to immunovirotherapy, a significant increase in survival

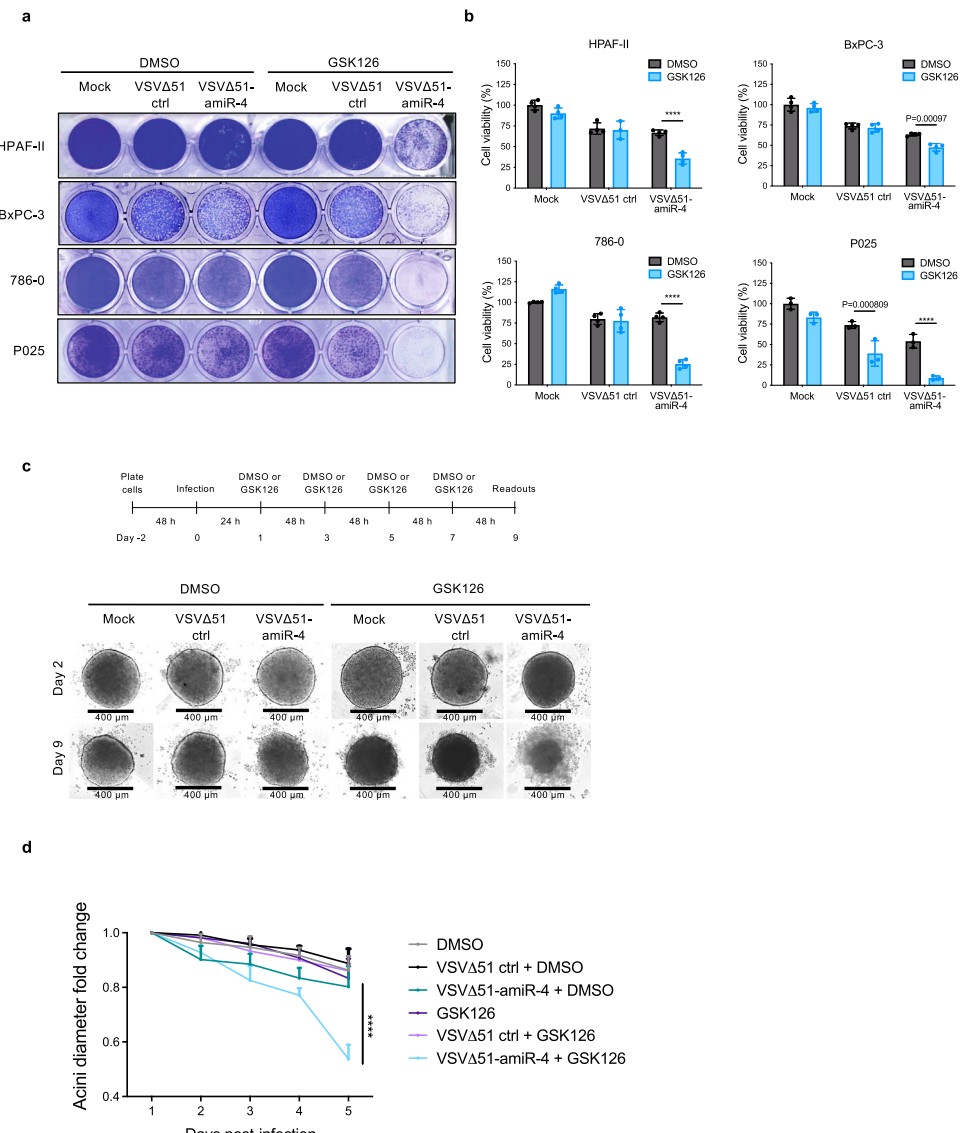

**Fig. 3 VSVΔ51-amiR-4 infection and inhibition of EZH2 via GSK126 promote synthetically lethal conditions and potentiate anti-tumour therapy.**
**a**, **b** When exposed to low viral doses of VSVΔ51-amiR-4 and 2–3 doses of GSK126 (15 μM) depending on the cell line, cell death is increased in a panel of cancer cell lines (HPAF-II, BxPC-3, 786-0) and a primary patient-derived pancreatic cancer cell line (P025), as assessed by representative crystal violet staining images of four biological replicates (**a**) and corresponding cell viability quantification compared to mock-infected DMSO-treated cells (**b**); timeline of treatment procedures can be found in Supplementary Fig. 3a. Two-way ANOVA with Sidak's multiple comparisons test (95% CI), ****$P < 0.0001$. Shown are the mean values ± SEM of biological triplicates (P025) or quadruplicates (HPAF-II, BxPC-3 and 786-0). **c** Phase-contrast images (scale bar = 400 μm) and (**d**) acini diameter fold change compared to the diameter at day 0 of uninfected or VSVΔ51-amiR-NTC or VSVΔ51-amiR-4-infected BxPC-3 spheroids with or without 15 μM GSK126 treatment. Results displayed as mean ± SEM, $n = 3$ biological replicates per condition. For day 5, two-way ANOVA with Dunnett's multiple comparisons test (95% CI), ****$P < 0.0001$. Source data are provided as a Source data file.

was evident in animals treated with VSVΔ51-amiR-4 compared to vehicle control ($P$ value = 0.0020) or VSVΔ51 control-treated mice ($P$ value = 0.0021) (Fig. 5e). Similarly, mice bearing ID8 Trp53$^{-/-}$ ovarian peritoneal tumours demonstrated increased survival upon treatment with VSVΔ51-amiR-4 compared to vehicle control ($P$ value = 0.0040) or VSVΔ51 control ($P$ value = 0.021) (Supplementary Fig. 5b).

Importantly, the combination of VSVΔ51-amiR-4 with GSK126 was more effective at treating mice harbouring human HPAF-II SC tumours compared to either the virus or drug alone (Fig. 5f and Supplementary Fig. 5c). Furthermore, the combination treatment regimen results in durable tumour remissions in both syngeneic murine B16-F10 orthotopic (Fig. 5g and Supplementary Fig. 5d) and B16-F10 peritoneal carcinomatosis

model (Supplementary Fig 5e). We found little change in the immunological profiles of tumours treated with VSVΔ51 control or VSVΔ51-amiR-4 (Supplementary Fig. 6a, b). We also did not detect significant changes in presence of oncolytic virus with or without combinatorial GSK126 treatment at both early time points (Supplementary Fig. 6c, d) and later time points (Supplementary Fig. 6e, Supplementary Data 2). These data suggest that the therapeutic impact of ARID1A downregulation by an OV platform and its combination with GSK126 does not significantly impact the immune landscape in the TME.

**VSVΔ51-amiR-4 enhances immune checkpoint inhibition.** Recent studies have shown that loss of *ARID1A* expression

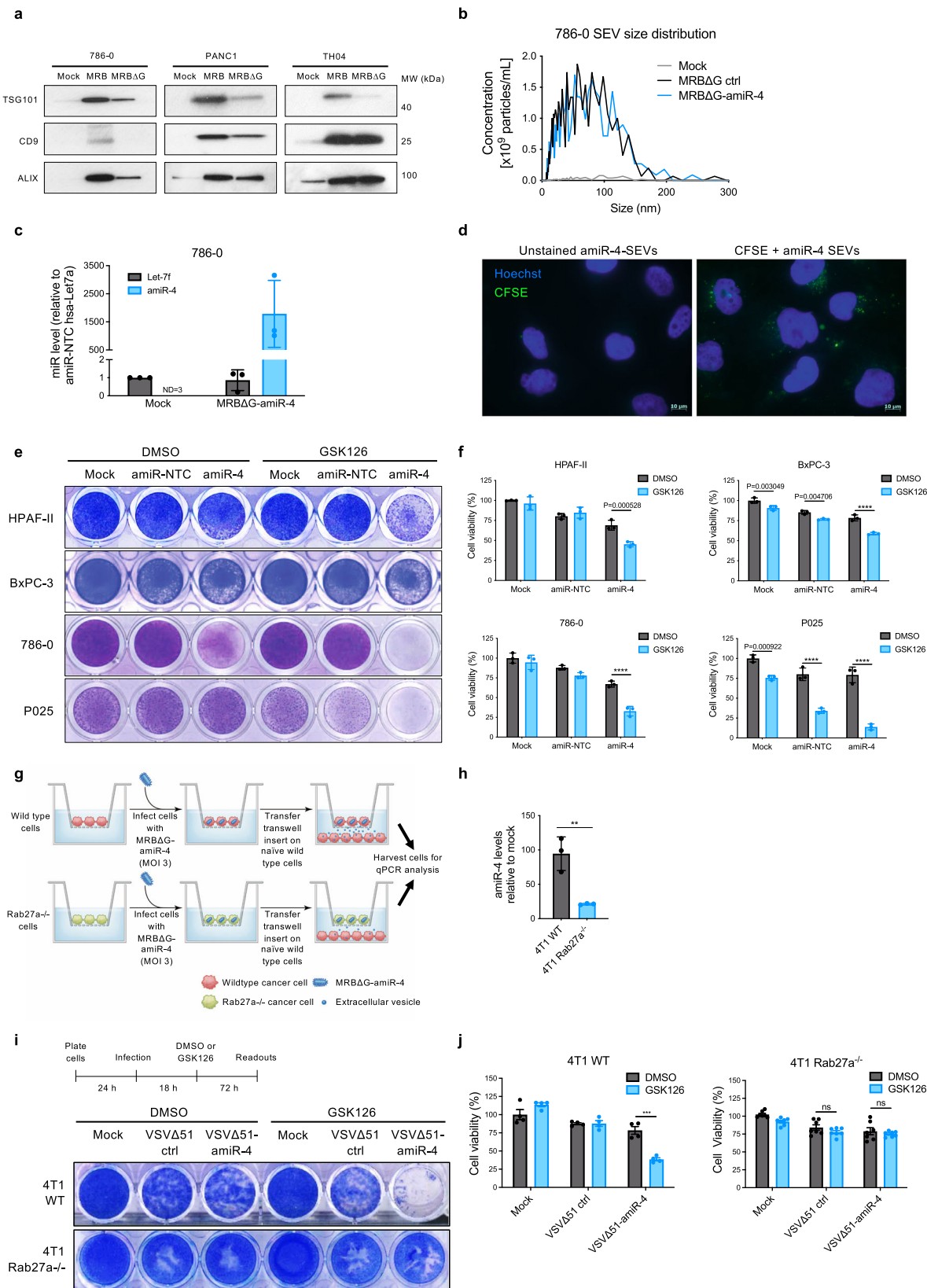

sensitises tumours to CTLA4 and PD-1 immune checkpoint inhibition[32,33]. Thus, we thought that VSVΔ51-amiR-4 therapeutic activity may benefit by further bolstering immune responses in the tumour with immune checkpoint inhibitor therapy. We next examined whether VSVΔ51-amiR-4 has better anti-tumour efficacy when combined with anti-CTLA4. VSVΔ51-

amiR-4 combined with anti-CTLA4 treatment significantly (P value = 0.0013) enhanced tumour regression and prolonged overall survival of mice bearing subcutaneous (SC) B16-F10 melanoma tumours (Supplementary Fig. 7a, b) and was superior to anti-CTLA4 therapy when combined with VSVΔ51 that lacks amiR-4.

**Fig. 4 Small extracellular vesicle-mediated transport of amiR-4 to uninfected cells contributes to enhanced cytotoxicity. a** Immunoblotting analysis of ALIX, CD9 and TSG101 in total purified SEVs produced by indicated cell lines with or without infection with MRB or MRBΔG (n = 3). **b** NTA showing size distribution and quantification of SEVs produced from mock-infected or MRBΔG-infected 786-0 cells (n = 3). **c** RT-qPCR analysis of amiR-4 levels compared to hsa-Let-7f-5p (loading control) in mock cell-associated SEVs and infected cell-associated SEVs derived from 786-0 cells infected with MRBΔG-amiR-4. Results displayed as mean ± SD, n = 3. ND not detected. **d** Representative immunofluorescence images showing uptake of CFSE-labelled SEVs derived from MRBΔG-amiR-4-infected 786-0 cells by naive 786-0 cells (n = 2). Nuclei were stained with Hoechst dye. Scale bar = 10 μm. **e**, **f** Indicated cancer cells were educated with SEVs harvested from mock-infected, MRBΔG-amiR-NTC or MRBΔG-amiR-4-infected cells and treated with vehicle control (DMSO) or GSK126. Representative crystal violet cell cytotoxicity assay images (**e**) and their corresponding cell viability quantification compared to mock SEV and DMSO-treated cells (**f**) are shown; the timeline of treatment procedures is shown in Supplementary Fig. 4g. Shown are means ± SD, n = 3. Two-way ANOVA with Sidak's multiple comparisons test (95% CI), ****P < 0.0001. **g** Schematic representation of transwell coculture assays designed to assess the transfer of amiR-4 via infected cell-derived SEVs to uninfected cells. **h** RT-qPCR analysis of amiR-4 levels in receiving 4T1 wild-type (WT) cells (lower compartment) after 48 h of education by cell-secreted factors derived from MRBΔG-amiR-4-infected cells (4T1 wild-type [WT] or Rab27a-deficient [Rab27a$^{-/-}$] cells; upper compartment). Results displayed as mean ± SD, n = 3, two-tailed unpaired t test, **P = 0.0066. **i**, **j** 4T1 WT (n = 4) or Rab27a$^{-/-}$ cells (n = 7) were mock-treated or infected with VSVΔ51 control or VSVΔ51-amiR-4 (MOI 0.025) and treated with either vehicle control (DMSO) or GSK126 (15 μM). Representative crystal violet cell cytotoxicity assay images (**i**) and their corresponding quantifications (**j**) are shown as mean ± SEM, two-way ANOVA with Tukey's multiple comparison test (95% CI), $^{ns}$P > 0.05, ***P = 0.0003. Source data are provided as a Source data file.

To assess the broader applicability of our approach, we also chose to rationally design and bioengineer an OV platform that expresses an artificial miRNA-based shRNA sequence against PD-L1 that will be packaged into infected cell-derived SEVs for delivery. In addition to being often expressed at high levels in cancer cells, PD-L1 expression is upregulated upon infection with many viruses[34,35] due to the secretion of immune factors in the infection site. In agreement with the results of other studies, we found that infection of cancer cells with VSVΔ51 induced a marked upregulation of PD-L1 (Fig. 6a). Expression of a shPD-L1 from a VSVΔ51 backbone decreased PD-L1 expression to basal levels (Fig. 6a) while not impacting virus growth or induced cytotoxicity (Supplementary Fig. 7c, d). We then cocultured B16-F10 cells infected with a replicating but non-spreading VSVΔG virus expressing shPD-L1 with uninfected cells (Fig. 6b). Immunoblot analysis of the recipient cells following the transwell assay showed specific downregulation of PD-L1 only in cells that were cocultured with VSVΔG-shPD-L1-infected donor cells (Fig. 6c). PD-L1 is an immune surface protein that inhibits anti-tumour functions of T cells by binding to its receptor PD-1 and effectively protects tumours from immune surveillance[36,37]. Finally, we examined the efficacy of VSVΔ51-shPD-L1 in vivo. Mice bearing orthotopic B16-F10 tumours were treated with either VSVΔ51-shNTC control or VSVΔ51-shPD-L1. VSVΔ51-shPD-L1 therapy was well-tolerated by animals, with no significant differences in body weight between treatment groups throughout the experimental period (Supplementary Fig. 7e). While VSVΔ51-shNTC has a moderate effect, VSVΔ51-shPD-L1 enables better tumour control and relapse-free rejection in 40% of the animals (Fig. 6d, e).

Lastly, we armed an OV with multiple artificial microRNAs with the ultimate goal of modulating the expression of different therapeutic gene targets in the tumour microenvironment. We constructed a single VSVΔ51 virus that expresses both an artificial microRNA directed against PD-L1 and a second microRNA (amiR-4) targeting the ARID1A transcript. We showed that when we infect cancer cell lines in vitro with this single virus, both gene products can be downregulated (Fig. 6f, g). These results suggest that OV platforms can be rationally bioengineered to target various molecules in the TME by in situ expressing multiple therapeutic artificial microRNA sequences.

## Discussion

Oncolytic viruses have been selected or engineered to take advantage of tumour-specific signalling defects. However, like all other cancer therapeutics, they are limited by the genetic and epigenetic heterogeneity that is inherent in any individual's malignancy. Here, we describe a strategy to broaden the landscape of tumours that can be effectively attacked by OV therapy. There is an extensive cellular signal transduction and effector network that has overlapping activities in the regulation of both cell growth and antiviral responses[10]. Genetic mutations or epigenetic modifications of the network that enhance cancer cell growth and survival as well as immune escape appear to simultaneously diminish antiviral responses, although the extent of this effect likely depends upon the specific type of mutation[38].

In earlier work, it was demonstrated that the pathological properties of a benign cytoplasmic virus could be enhanced by encoding amiRNAs that downregulate the expression of key antiviral cellular proteins[39]. We hypothesised that there also exists a subset of cellular targets that could be downregulated by OV-driven expression of an amiRNA to specifically sensitise cancer cells while leaving normal cells unharmed. This idea was based upon the concept of synthetic lethality that is being exploited for the development of anticancer small molecules[40,41]. We reasoned that a "second-hit" in a cancer cell could make it hypersensitive to virus infection, but not impact virus growth in normal tissues. In the surrogate SV-amiRNA library screen, two amiRNAs were found to confer a replicative advantage to SV in normal cells, notably amiR-1 and amiR-2 (Fig. 1b), and thus were not appropriate for the therapeutic strategy described here. On the other hand, amiR-4, when encoded in an oncolytic rhabdovirus, provided no replicative advantage in healthy human GM38 cells (Supplementary Fig. 1f) but enhanced virus growth and killing in a variety of tumour cell lines (Fig. 1e and Supplementary Fig. 1c–e, g).

Recently natural miRNAs encoded in an oncolytic adenovirus have been shown to enhance the therapeutic activity of the vector in a cell-autonomous fashion[42]. Many different OV platforms may potentially be combined with our miRNA technology, but it will be necessary to consider the unique biology of each virus family, as well as the susceptibility of individual tumours to the different OVs. In this study, we identify *ARID1A*, *PLEC* and *HDAC4* as predicted amiR-4 targets and show reduced gene expression and protein levels following VSVΔ51-amiR-4 infection (Fig. 2a, b). In line with previous reports, these proteins are involved in epigenetic remodelling and cytoskeleton stability, cellular processes that can be chemically manipulated to enhance OV efficacy[19,43]. Our RNA-seq analysis revealed a dampening of interferon response elements in *ARID1A*-knockout cells (Fig. 2k, i) consistent with a previously suggested role of ARID1A in the regulation of the interferon response[44]. Indeed, depleting cancer

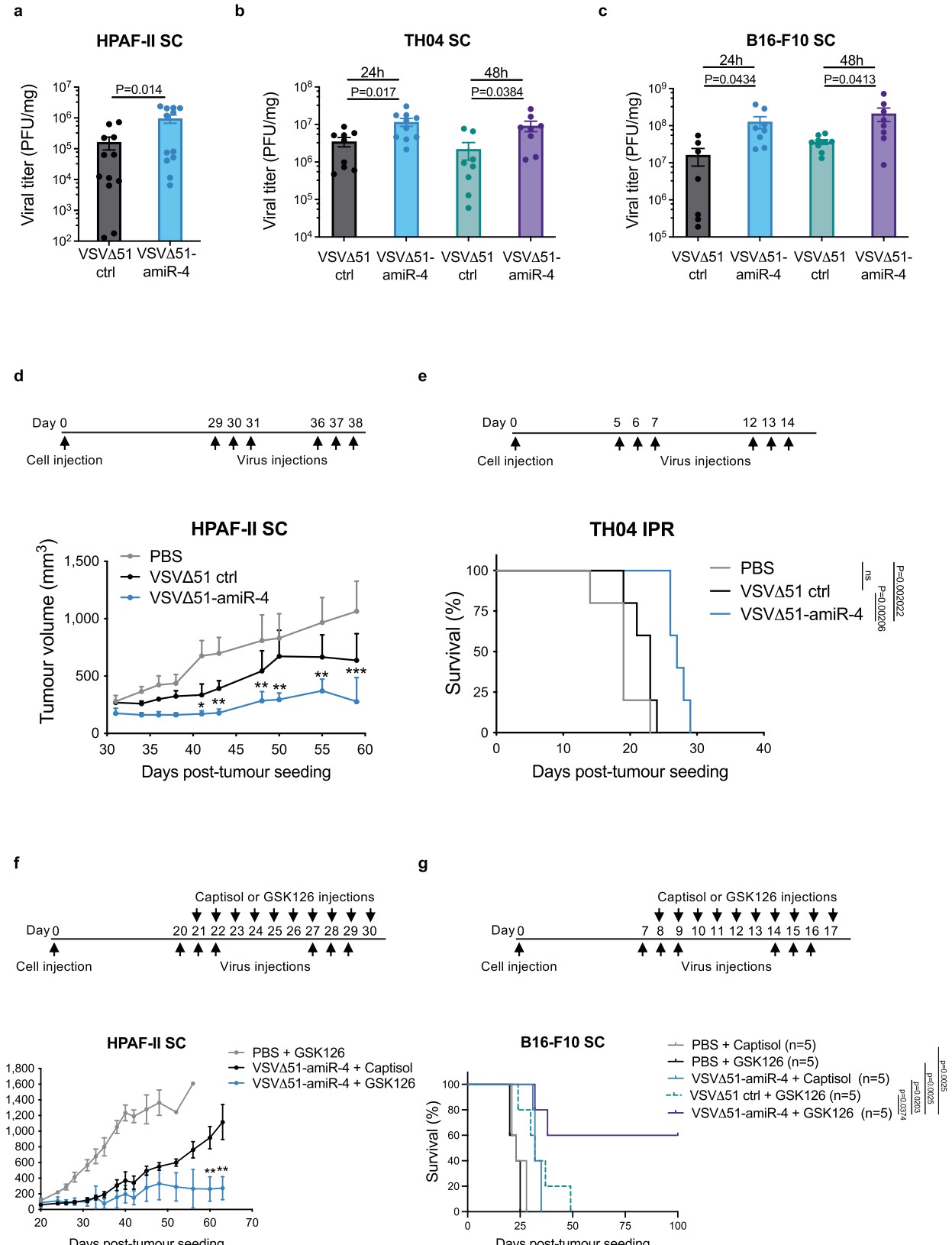

cells of ARID1A using CRISPR-Cas9 technology or by infecting *ARID1A* wild-type cells with VSVΔ51-amiR-4 renders these cells significantly more sensitive to infection and virus-mediated killing (Fig. 2c and Supplementary Fig. 2c). Interestingly, our RNA-seq analysis identified IFITM2 as the protein most significantly downregulated in *ARID1A*-knockout cells compared to wild-type

cells (Fig. 2i). Previous reports demonstrated a role of IFITM2 in restricting VSV infection in HEK293T cells with increased viral titers upon treatment of cells with a siRNA against IFITM2[45].

We describe here an amiRNA strategy to enhance oncolytic virus spread in cancer cells through SEV production and spread. In addition, we demonstrated that this strategy can equally be used to

**Fig. 5 amiR-4 enhances oncolytic virus replication and survival of tumour-bearing xenograft and immunocompetent murine models in combination with GSK126. a–c** A single dose of VSVΔ51 expressing amiR-NTC or amiR-4 was delivered IT into SC HPAF-II ($n = 12$ per group), TH04 (24 h: $n = 9$ for VSVΔ51 ctrl, $n = 10$ for VSVΔ51-amir-4; 48 h: $n = 8$ per group) or B16-F10 (24 h: $n = 7$ for VSVΔ51 ctrl, $n = 8$ VSVΔ51-amir-4; 48 h: $n = 9$ for VSVΔ51 ctrl, $n = 8$ for VSVΔ51-amir4) tumours. After 24 h (**b**, **c**) or 48 h (**a–c**) post delivery, virus titers were quantified. Data represent mean values ± SEM of two pooled independent experiments. Unpaired two-tailed $t$ test. **d** Six doses of PBS ($n = 4$), VSVΔ51-amiR-NTC ($n = 3$) or VSVΔ51-amiR-4 ($n = 4$) were delivered IT into mice bearing SC HPAF-II tumours. Mean tumour volumes ± SEM are shown ($n = 5$ per group). Two-way ANOVA with Tukey's multiple comparison test (95% CI), *$P < 0.05$, **$P < 0.01$, ***$P < 0.001$. Exact $P$ values are provided in the Source data file. **e** Kaplan–Meier survival analysis of orthotopic (intrapancreatic (IPR)) TH04 pancreatic tumour-bearing mice treated with six IP doses of PBS, VSVΔ51-amiR-NTC or VSVΔ51-amiR-4. Log-rank (Mantel–Cox) test, ($n = 5$ per group), $^{ns}P > 0.05$. **f, g** Tumour-bearing mice were treated as indicated with vehicle controls (PBS and/or Captisol) or with VSVΔ51 control or VSVΔ51-amiR-4 or GSK126 (50 mg/kg) or the combination of both monotherapies. **f** Individual tumour growth curves of mice with SC HPAF-II tumours after indicated treatments. Tumour volumes are displayed as mean ± SEM (vehicle controls, $n = 5$; VSVΔ51-amiR-4 alone or with GSK126, $n = 4$). Days 60 and 63: Two-way ANOVA with Sidak's multiple comparisons test (95% CI), **$P < 0.01$. Exact $P$ values are provided in the Source data file. **g** Kaplan–Meier survival curves of mice bearing B16-F10 tumours and Log-rank (Mantel–Cox) test ($n = 5$ mice per group). Source data are provided as a Source data file.

target other cellular components of the TME, including immune surface molecules, which are often upregulated in malignant cells and function to protect tumours from immune surveillance (Fig. 6a–g). This suggests an attractive strategy, as small genetic packages, like optimised amiRNAs, could be readily used to downregulate challenging or "undruggable" therapeutic targets within the TME. As an example, there are no known drugs that target ARID1A, but through SEV delivery of amiR-4 to uninfected cells, we created a synthetic lethal phenotype that can be exploited through the inhibition of EZH2 by GSK126 (Fig. 5f, g and Supplementary Fig. 5c–e). A second benefit of targeting ARID1A is the recent demonstration that inactivation of certain SWI/SNF chromatin remodelling complex subunits (i.e. *ARID1A* and *PBRM1*) renders tumours more likely to respond to ICIs[32,46]. Notably, a combination of VSVΔ51-amiR-4 with anti-CTLA4 antibodies enabled profound tumour control and relapse-free rejection in 60% of the mice (Supplementary Fig. 7a, b).

Together, our data demonstrate that VSVΔ51-amiR-4-infected cells induce a bystander effect in the TME by secreting amiR-4-containing SEVs which facilitate tumour cell death via a synthetic lethal interaction between ARID1A and EZH2 with the addition of the small-molecule inhibitor GSK126 (Fig. 6g). We showed that this approach can also be used to customise SEVs to transfer amiRNAs targeting immune checkpoint proteins such as PD-L1, which are often upregulated in cancer cells and block the anti-tumour function of T cells (Fig. 6g). Furthermore, we designed an oncolytic virus whereby can simultaneously downregulate ARID1A and PD-L1 by multiplexed miRNA technology (Fig. 6f) with the goal of demonstrating that OV can be designed to regulate multiple gene targets in the TME (Fig. 6g). While the development of SEV therapeutics and small RNA therapy is advancing, there are still limitations associated with their manufacture and selective delivery. The strategy described here overcomes these hurdles by facilitating both the in situ manufacturing and localised delivery of therapeutic SEVs within tumour beds.

## Methods

The research presented herein complies with all relevant ethical regulations at OHRI and the University of Ottawa (biohazardous material use certificate GC317-125-12). All patient samples were obtained through the Global Tissue Consenting committee at the OHRI (pancreatic patient tumour-derived collection protocol 20120112-01H). All animal studies were approved by the institutional animal care committee of the University of Ottawa (Protocol ID: OHRI2870 and MEe-2258) and carried out in accordance with guidelines of the National Institutes of Health and the Canadian Council on animal care.

**Reagents**. Unless otherwise stated, all reagents were purchased from Sigma Chemical Co. (St. Louis, MO).

**Cells and cell culture conditions**. MIA PaCa-2 (CRM-CRL-1420), 786-0 (CRL-1932), Vero (CCL-81), BxPC-3 (CRL-1687), PANC1 (CRL-1469), HPAF-II (CRL-

1997), 4T1 (CRL-2539), B16-F10 (CRL-6475), HPAC (CRL-2119), SKOV3 (HTB-77) and BHK-21 (CCL-21) cells were purchased from the American Type Culture Collection (ATCC, Manassas, VA). Normal human GM00038 skin fibroblasts (abbreviated herein as GM38, NIGMS Human Genetic Cell Repository), human foetal pancreatic fibroblasts (SC00A5, herein named as PCa-CAF), and T-REx™-293 cells (R71007) were obtained from the Coriell Institute Cell Repositories (Camden, NJ), Vitro Biopharma (Golden, CO), and Invitrogen, respectively. The murine pancreatic adenocarcinoma cell line Panc02 (CVCL_D627; NCI-DTP Repository) of C57BL/6 origin was developed by Corbett et al.[47]. Mel888 cells[48] were a gift from Dr. Melcher at the Institute of Cancer Research, London, UK. 786-0 and PANC1 CRISPR-Cas9 mediated ARID1A-knockout and 4T1 Rab27a knockout cell lines were generated using the lentiCRISPRv2 protocol as described below. The primary P025 cells were freshly derived from a dissociated pancreatic cancer patient-derived xenograft. TH04 murine pancreatic cancer cells[49] were a gift from Dr. Juliana Candido at the Barts Cancer Institute, Queen Mary University of London, UK. The TH04 cell line was derived from spontaneous pancreatic tumours of Kras (G12D) and Trp53 (R172H) C57BL/6 mice. ID8 *Trp53*$^{-/-}$ cells[50] were a gift of Dr. Iain McNeish (Barts Cancer Institute, Queen Mary University of London, UK). MIA PaCa-2, 786-0, Vero, BxPC-3, PANC1, Panc02, HPAF-II, 4T1, B16-F10, SKOV3, BHK-21, PCa-CAF, TH04, ID8 *Trp53*$^{-/-}$ and Mel888 cells were cultured in Dulbecco's minimal essential medium (DMEM; Corning) containing 10% foetal bovine serum (FBS; Hyclone). 786-0 ARID1A$^{-/-}$, PANC1 ARID1A$^{-/-}$, 4T1 Rab27a$^{-/-}$ were cultured in DMEM containing 10% FBS and puromycin (3.5 µg/ml for 786-0 ARID1A$^{-/-}$, 4 µg/ml for PANC1 ARID1A$^{-/-}$, 1 µg/ml for 4T1 Rab27a$^{-/-}$; Cayman Chemical). GM38 fibroblasts were cultured in DMEM containing 2% FBS. T-REx™-293 cells were cultured in DMEM containing 10% FBS, zeocin (300 µg/ml; Gibco) and blasticidin (5 µg/ml; Invivogen). HPAC cells were grown in RPMI-1640 medium (Corning), 10% FBS, 15 mM HEPES, 0.5 mM sodium pyruvate, 2 µg/mL insulin, 5 µg/ml transferrin, 40 mg/ml hydrocortisone, 10 mg/ml hEGF. P025 cells were grown in RPMI-1640 medium, 10% FBS, 15 mM HEPES, 0.5 mM sodium pyruvate, 2 µg/ml insulin, 5 µg/ml transferrin, 40 mg/ml hydrocortisone, 10 mg/ml hEGF, 100 µg/mL Gentamicin, 0.1 mg/ml Normocin™ (InvivoGen) and 1× antibiotic–antimycotic (Gibco). All cell lines were cultured under 5% $CO_2$ at 37 °C and were routinely tested for mycoplasma contamination using Hoechst stain (Invitrogen) and the e-Myco VALiD Myco PCR detection kit (FroggaBio). All cell lines have tested negative for mycoplasma contamination and all cell lines were used within passage no. 3 and passage no. 30.

**Generation of CRISPR knockout cell lines**. The lentiCRISPRv2 plasmid was digested with FastDigest BbsI (NEB) and gel purified using the QIAquick Gel Extraction kit (Qiagen). Forward and reverse gRNA oligonucleotides coding for *ARID1A* or *Rab27a* were phosphorylated and annealed according to the Zhang lab lentiCRSPRv2 and lentiGuide oligo cloning protocol. Lenti-CRISPR v2 was a gift from Feng Zhang [Addgene plasmid # 52961; http://n2t.net/addgene:52961]$^{51}$ RRID:Addgene_52961][51]. Annealed oligonucleotides (Integrated DNA Technology) were diluted 1:200 and ligated into BbsI digested lentiCRISPRv2 at room temperature with Quick Ligase (NEB) and then transformed into Stbl3 cells (Invitrogen). DNA was extracted using the Qiagen MiniPrep kit (Qiagen) and verified by Sanger sequencing. gRNA targets were chosen from the GeCKO Lentiviral sgRNA v2 libraries; hARID1A g1: 5′-GATGCATGATGCTGTCCGAC-3′, mRab27a g1: 5′-GTTTCCTCAATGTCCGAAAC-3′. A lentivirus encoding the individual gRNAs was produced using 3rd generation packaging plasmids[52]. Cells were seeded in six-well plates at a low density such that they were 40–50% confluent at the time of transduction. Cells were transduced with 1 ml of lentivirus and 24 h post-transduction, were placed in selective media containing puromycin (4 µg/ml for PANC1 ARID1A$^{-/-}$, 3.5 µg/ml for 786-0 ARID1A$^{-/-}$ and 1 µg/ml for 4T1 Rab27a$^{-/-}$). Once the selection was complete, limiting dilution plating was performed in order to obtain single-cell colonies. Clonal cell lines were expanded such that lysates and DNA could be harvested to assess target gene knockout. Targeted knockout was

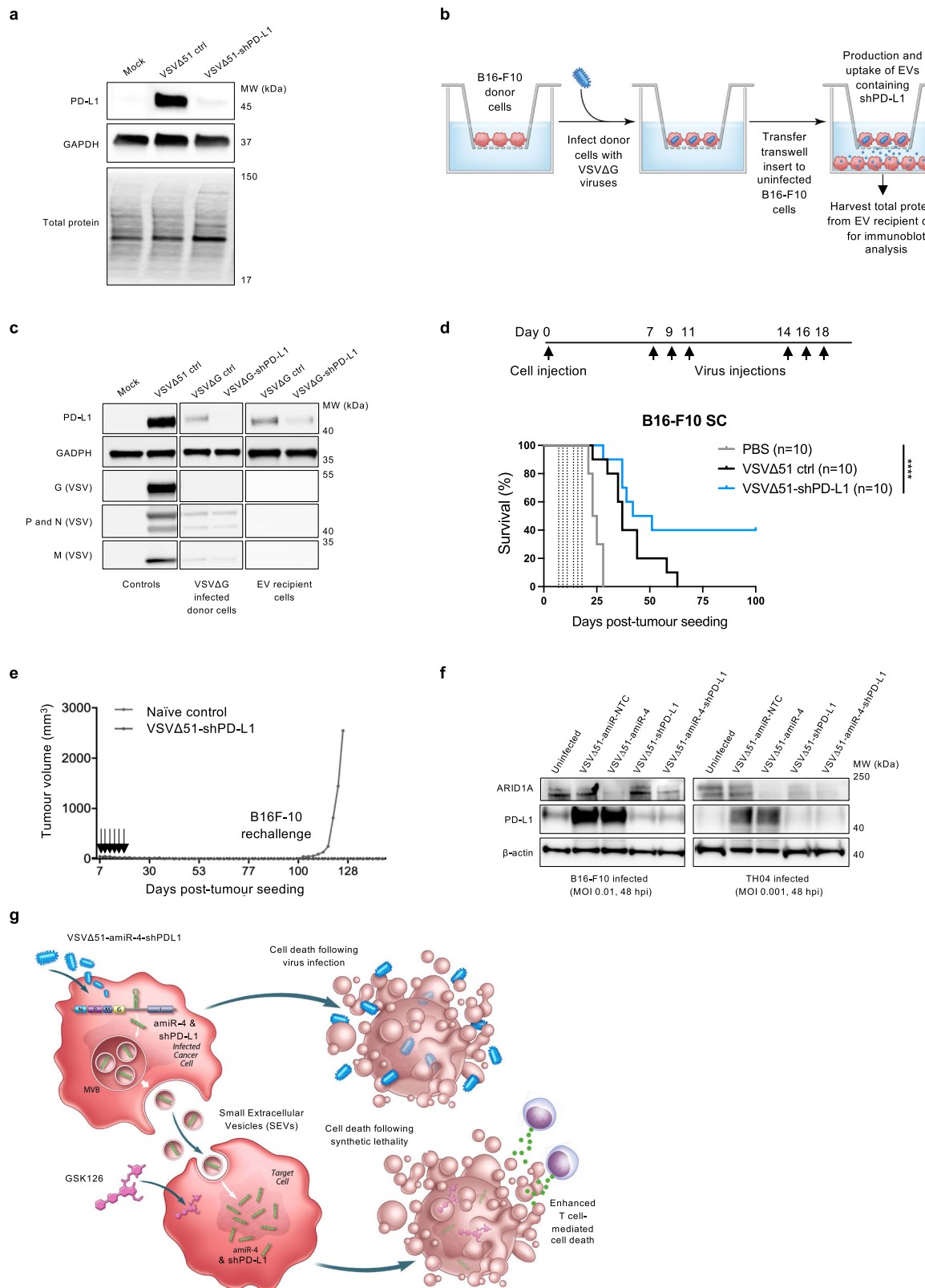

confirmed using western blotting, T7 endonuclease assay and Sanger sequencing. T7 endonuclease assay to confirm gRNA cleavage was performed using DNA from clonally derived cell lines (PANC1, 786-0 and 4T1) was extracted using the DNeasy Blood & Tissue kit (Qiagen). Genomic DNA was used as the template to amplify the targeted locus of interest using Q5 Hot Start High-Fidelity 2× Master Mix (NEB) and following the manufacturer's protocol. Primer pairs are listed in Supplementary Table 4. PCR products were purified using the PureLink PCR Purification kit (Thermo Fisher). Purified DNA was quantified by Nanodrop, and 200 ng of DNA was mixed with 10× NEBuffer 2 and dH$_2$O to a final volume

of 19 μl. Samples were heated to 95 °C for 5 min, followed by a slow ramp-down to room temperature to enable heteroduplex formation. T7 endonuclease I (1 μl; NEB) was added to each sample and incubated at 37 °C for 15 min. Immediately following digestions, samples were resolved on a 2% agarose gel to assess for the presence of Indels. Clonally derived PANC1, 786-0 and 4T1 cell lines with a positive T7 result were sent for Sanger sequencing to confirm T7 results and determine Indel size and location. All sequencing was completed at StemCore Laboratories (Ottawa, ON) using the primers (IDT) listed in Supplementary Table 4.

**Fig. 6 An OV targeting PD-L1 via extracellular vesicle delivery of shPD-L1 molecules has enhanced therapeutic activity in vivo. a** Immunoblot analysis showing protein expression levels of PD-L1, GAPDH (loading control), and total proteins in B16-F10 cells subjected or not to VSVΔ51-amiR-NTC or VSVΔ51-shPD-L1 infection ($n = 3$). **b**, **c** Schematic representation (**b**) of transwell coculture assays designed to assess the transfer of shPD-L1 via infected cell-derived SEVs to uninfected cells. **c** Immunoblot analysis of PD-L1, GAPDH and VSV protein levels in producer cells (upper compartment) and receiving cells (lower compartment) after 48 h of education by cell-secreted factors derived from VSVΔG-shPD-L1-infected cells (upper compartment). Both uninfected and VSVΔ51-infected cell lysates are included as controls. Immunoblots are representative of three biological replicates. **d** Kaplan–Meier survival curves of mice bearing subcutaneous (SC) B16-F10 tumours and treated as indicated with vehicle control (PBS) or with VSVΔ51-amiR-NTC control or VSVΔ51-shPD-L1 ($n = 10$ mice per group). Dotted vertical lines indicate virus treatments. Log-rank (Mantel–Cox) test, ****$P < 0.0001$. **e** Mice that had completely cleared B16-F10 tumours upon VSVΔ51-shPD-L1 treatment (in **d**) were re-challenged SC with $5 \times 10^5$ B16-F10 cells on day 93 ($n = 4$). Arrows indicate initial VSVΔ51-shPD-L1 treatments. **f** B16-F10 and TH04 cells were infected with indicated OVs (VSVΔ51-amiR-NTC, VSVΔ51-shPD-L1, VSVΔ51-amiR-4, VSVΔ51-amiR-4-shPD-L1) at MOIs of 0.01 and 0.001, respectively. Cell lysates were collected 48 hpi and prepared for immunoblotting with antibodies against ARID1A, PD-L1, and β-actin (loading control), $n = 3$. **g** Schematic showing the bystander effects of VSVΔ51-amiR-4-shPD-L1. In our model, cell death occurs in a two-pronged attack against cancer cells; following OV infection of cancer cells and via synthetic lethal interactions in target cells receiving ARID1A-targeting amiR-4 delivered by SEVs from neighbouring OV-infected cells and systemic administration of the EZH2 methyltransferase inhibitor GSK126. In addition, another bystander effect is induced by shPD-L1-containing SEVs derived from VSVΔ51-shPD-L1-infected cells. While OV infection induces PD-L1 protein levels, expression of an shRNA targeting PD-L1 from an OV backbone decreases PD-L1 to baseline levels and enhances T cell-mediated death of cancer cells. Source data are provided as a Source data file.

**Viruses**. The oncolytic VSVΔ51[16], Maraba[17], oHSV-1[53] and VV TK⁻ VGF⁻ [54] virus backbones and propagation protocols have previously been described. In brief, to construct and rescue the replication-competent amiRNA-expressing VSVΔ51 viruses, plasmids containing the amiR-1-5 sequences or a shRNA against mouse PD-L1 or the non-targeting control (NTC; sequence targeting the GFP mRNA) (Supplementary Table 1) encoded in the pre-miR-30-based short hairpin cassette (Supplementary Fig. 1a) flanked with XhoI and NheI restriction sites were purchased from GenScript. Both amiR-encoding plasmids and VSVΔ51-GFP encoding plasmids were digested using XhoI and NheI (NEB) and amiRNA inserts were ligated individually into the VSVΔ51 empty vector at the gene junction between G and L proteins as previously described[55]. Single-cycle MRBΔG viruses (replication-defective) were generated using a similar strategy by first PCR amplifying the corresponding pre-amiR sequences from GenScript plasmids or the NTC (sequence targeting the Firefly luciferase mRNA) using specific primers containing the BsrD1 and KpnI restriction enzyme sites (forward primer: 5′-GCAATGACGAGTTTGTTTGAATGAGGCTTC-3′; reverse primer: 5′-GGTACC AAAGTGATTTAATTTATACCATTTTA-3′; Integrated DNA Technology). A plasmid containing the Maraba virus genome where the glycoprotein (G) gene was replaced by the GFP sequence (pBR-MRBΔG) was then used to insert the desired pre-miR cassette. Briefly, the pBR-MRBΔG vector was digested with BsrD1 and KpnI to remove the GFP gene, and then the digested vector and PCR-amplified pre-miR cassettes were ligated. The resulting plasmid contains the MRBΔG genome and the amiR sequences, which were inserted between the M and L genes. All constructs were verified by Sanger sequencing (StemCore Laboratories, Ottawa, ON). All viruses were rescued using an infection-transfection method as previously described[55]. In the case of single-cycle viruses, MRBΔG and VSVΔG, viruses were rescued and titrated in T-REx™-293 cells expressing the G protein. T-REx™-293 cells containing a plasmid with doxycycline-inducible VSV G protein expression were used to produce the MRBΔG virus stocks. G protein expression was first induced with 800 μM doxycycline (DOX) and cells were immediately infected at MOI 0.05 in serum-free DMEM. After 2 h, the inoculum was removed and fresh DMEM with 10% FBS was added. After 48 h, media were harvested, and large cellular debris was removed using a $2000 \times g$ spin for 15 min at 4 °C. The media was further cleared by filtering on a 0.22 μm Steritop™ filter unit (Millipore). Virus was pelleted at $71,000 \times g$ for 2 h (Beckman Coulter Avanti J-E centrifuge).

**Sindbis virus amiRNA library passaging**. A replicating Sindbis virus amiRNA library (SV-amiRNA library) encoding ~16,000 unique amiRNA sequences was used in our studies. The targeting amiRNA sequences in our library were designed using the Hannon and Elledge algorithm [http://www.ncbi.nlm.nih.gov/projects/genome/probe/doc/DistrOpenBiosystems.shtml]. Essentially, a pre-miR-30 cassette is used to house the shRNA and approximately 16,000 unique amiRNAs were originally cloned into a Sindbis virus expression vector. This library has been previously described[39]. To serially passage the library in malignant and healthy cells, we first infected PANC1, MIA PaCa-2, Panc02, PCa-CAF and GM38 normal fibroblasts with the original pooled SV-amiRNA library at a MOI of 0.1. 24 hpi, virus outputs were collected and saved for subsequent infection cycles. A fraction of the sample was saved separately and tittered by plaque assay in Vero cells. Four consecutive serial passages of the SV-library were conducted using five parallel biological replicates (a schematic representation of our screening pipeline can be found in Fig. 1a).

**Targeted RNA sequencing of the passaged SV-amiRNA library**. Targeted RNA sequencing from both the unpassaged (input) and passaged library samples was performed from total RNA collected using an RNeasy Kit (Qiagen), as previously described[39]. Input samples were collected 18 hpi and passaged samples were collected

18 h following the fourth passage. Briefly, to monitor SV-amiRNA populations (diversity) upon passaging in selected healthy and malignant cells, total RNA was converted to cDNA using a SuperScript III reverse transcriptase (Invitrogen) and random hexamers. Specific primers for the pre-miR-30 cassette (Supplementary Table 1) with added Illumina-specific overhang adapter sequences compatible with high-throughput (HT) sequencing were used to amplify the amiRNA hairpin region. Deep sequencing analysis was performed using a MiSeq instrument (Illumina; Seq-Matic LLC, CA). As previously described and using custom shell and Python scripts[14], the data was stripped of adapters and barcodes and matched against the full list of 16,000 amiRNAs in the original library, generating observed counts for each amiRNA in each replicate. To control for sequencing errors/variations, the sequences were aligned and grouped in "families of sequences". Enrichment of specific amiRNA hairpins was calculated by comparing the frequency of each amiRNA in the input original library and in the passaged samples. The top five amiRNA candidates were selected for future analysis.

**RNA sequencing of PANC1 ARID1A wild-type and knockout clones**. PANC1 cells (5E5) were seeded into six-well plates. After 18 h, cells were infected with VSVΔ51 at MOI 3 or mock-infected with serum-free media. One hour after infection, media was aspirated and replaced with fresh media containing 10% FBS. RNA was harvested in TRIzol™ (Thermo Fisher) 18 hpi. RNA was extracted using the manufacturer's protocol and quantified by nanodrop. PANC1 wild-type or ARID1A-knockout total RNA was sequenced by The Centre for Applied Genomics (TCAG) at The Hospital for Sick Children (Toronto, Canada) using the Illumina HiSeq2500 platform to generate single-end 100 bp reads. Adapters were trimmed using Trimmomatic[56] and adapter-free reads were mapped to the human genome (hg19) using TopHat[57]. Transcript abundance and differential expression analysis were performed using cufflinks and cuffdiff[58] as part of a previously described pipeline[59]. Genes with an FDR value smaller than 0.05 and with a fold change greater or smaller than four were considered for pathway analysis using gProfiler[60]. Heatmaps were generated using Package "pheatmap" - R Project version 1.0.8.

**Bioinformatic prediction of amiR-4 targets**. Bioinformatic prediction of amiR-4 targets carried out by TargetS based on Total Delta Energy of Duplex or by BLAST complementarity revealed nine potential amiR-4 targets. *ARID1A*, *PLEC*, *HDAC4* and *MCM2* were selected for further validation based on their high targeting prediction values. From the initial nine targets predicted, three were significantly downregulated by amiR-4 (*ARID1A*, *PLEC* and *HDAC4*). Although *MCM2* was a predicted target, its downregulation was not observed in our in vitro systems and was thus used as a negative control. The remaining five targets were not significantly downregulated by amiR-4 in our in vitro systems.

**Quantitative real-time PCR**. Quantitative RT-PCR (RT-qPCR) was performed on non-pooled replicate samples. RNA extractions were performed using TRIzol™ reagent as per manufacturer protocol (Invitrogen). Cellular RNA was converted to cDNA by Superscript RT II (Invitrogen) or iScript™ cDNA synthesis kit (Bio-Rad) and EV miRNA was converted to cDNA using the microRNA cDNA synthesis kit (Quanta Bioscience). RT-qPCR was carried out using SYBR Green (Invitrogen) according to the manufacturer's instructions. Analysis was performed on a Rotor-Gene RG-3000A machine and the Rotor-Gene 6.1.81 software (Corbett Research) or a 7500 Fast Real-Time PCR System and the 7500 Software v2.3 (Applied Biosystems) according to the manufacturer's recommended protocols. Primer pairs specific for various gene products or miRNA sequences used in our experiments are provided in Supplementary Table 5. qRT-PCR measurements were normalised to the human *Rplp0* gene or Let7f-

1 miRNA using the $2^{-\Delta\Delta Ct}$ method[61]. Copy number per reaction in Supplementary Fig. 1b was calculated using the diluted sample in a three-point standard curve.

**Immunoblotting analysis.** Cells or SEVs were harvested and lysed in NP-40 buffer (1% NP-40, 150 mM NaCl, 2 mM EDTA, 50 mM Tris, pH 7.4) containing Complete™ EDTA-free protease inhibitors (Roche). Cell lysates were clarified by centrifugation at $12,000 \times g$ for 20 min at 4 °C. Proteins were separated on Nupage® 4–12% Bis-Tris Protein Gels (Invitrogen) and transferred on polyvinylidene fluoride (PVDF) membranes (Immobilon-P Millipore, Bedford, MA) for 2 h or overnight at 4 °C. Blocked membranes were incubated overnight at 4 °C with the following diluted antibodies: ALIX (1:2000; Santa Cruz Biotechnology, sc-53538), CD9 (1:1000; Abcam, ab236630), TSG101 (1:1000; Abcam, ab125011), Calreticulin (1:1000; BioVision, #3076), ARID1A (1:500; Cell Signaling Technology, #12354), HDAC4 (1:500; Abcam, ab12172), PLEC (Plectin-1) (1:500; Cell Signaling Technology, #2863), MCM2 (1:2000; Abcam, ab4461), GAPDH (1:1000; Abcam, ab37168 or 1:1000; Cell Signaling Technology, #2118), β-tubulin (1:1000; Cell Signaling Technology #2146), PD-L1 (1:1000; Abcam, ab213480); VSV (1:1000)[62] and Rab27a (1:500; Abcam, ab55667). After three washes with Tris-Buffered-Saline-Tween (TBS-T), the membranes were incubated with goat anti-rabbit (111-005-003) or goat anti-mouse (115-005-146) horseradish peroxidase-conjugated IgG (Jackson ImmunoResearch) for 1 h. All secondary antibodies were diluted 1:5000 in 5% (w/v) skim milk/TBS-T or bovine serum albumin (BSA) according to the manufacturer's recommendations. Membranes were washed three times with TBS-T and immunoreactive proteins were detected using Amersham ECL Western Blotting Detection Reagent (GE Healthcare) or Clarity ECL (Bio-Rad Laboratories) followed by exposure to X-ray film (Fuji Photo Film Co, LTD). Protein level quantification was assessed by densitometry analysis using ImageJ 1.52k software and was normalised to respective loading control expression levels.

**Cell viability assay.** Cell viability was assessed using the alamarBlue® Assay with the REDOX indicator resazurin according to the manufacturer's protocol. Briefly, media was removed, cells were washed once with 1× Dulbecco's phosphate-buffered saline (PBS; Corning) and fresh media was added to cells. Resazurin (2.5 mM) was added in a 1:10 dilution. Cells were incubated with resazurin at 37 °C for ~45 min to 3 h depending on the cell line. Fluorescence was measured (excitation 530 nm, emission 590 nm) on the Fluoroskan Ascent FL plate reader using the Ascent Software version 2.6. (Thermo Fischer Scientific).

**Crystal violet cytotoxicity assay.** Cytotoxicity was assessed using a crystal violet assay as previously described[63]. At the end of a given experiment, media was removed, cells were washed once with 1× PBS and a solution of 0.5% crystal violet was added to the cells. Plates were incubated at room temperature on a shaker for 30 min, then crystal violet solution was washed three times with water and plates were left to air-dry overnight. Once dry, plates were scanned to obtain pictures and were then subjected to crystal violet lift with 500 μl of methanol per well for a 24-well plate or with 250 μl of methanol per well for a 48-well plate. Plates were incubated at room temperature on a shaker for one hour to allow lifting of the crystal violet, then the $OD_{570}$ was read using the Multiskan Ascent plate reader using the Ascent Software version 2.6. (Thermo Fischer Scientific).

**Pancreatic cancer patient-derived xenograft samples.** Patient samples were obtained by endoscopic ultrasound-guided fine-needle aspiration biopsies (EUS-FNA) following protocol 20120112-01H approved by the research ethics board of the Ottawa Hospital Research Institute and with prior patient's informed consent. All samples were obtained following informed patient consent. Our work complies with ethical regulations set by our institution. Samples implanted subcutaneously in NOD-SCID mice (female, 6-8 weeks old, NOD.CB17-*Prkdc^scid*/NCrCrl, Charles River Laboratories, Wilmington, MA) were harvested once the tumour reached endpoint and subsequently used for histological analysis, cored for further experimental analyses, or used to establish primary cell cultures (P025). Tumour cores (2 × 2 mm) were infected with 1E5 plaque-forming units (PFUs) of VSVΔ51-amiR-NTC or VSVΔ51-amiR-4 for 48 h and 100 μl of the homogenised core samples were tittered by plaque assay as described below.

***ARID1A* gene mutation or deep deletion analysis in large pancreas cancer datasets.** We used the cBioPortal for cancer genomics online platform [https://www.cbioportal.org/] to query the indicated publicly available pancreatic cancer sequencing datasets and generate the graph displayed in Supplementary Fig. 2h.

**Tissue processing and immunohistochemistry.** Tumours were formalin-fixed, paraffin-embedded and sectioned before being subjected to IHC staining. After deparaffinization and rehydration of tumour block sections, antigen retrieval was performed in boiling sodium citrate buffer (pH 6.0). Tumour sections were stained for ARID1A (1:300; Abcam, ab182561).

**Plaque assays.** Titration of Rhabdoviruses: Samples containing rhabdoviruses were serially diluted and titered on Vero cells as previously described[55]. Briefly, a confluent monolayer of Vero cells was infected with tenfold serial dilutions of virus-containing samples for 1 h. Cells were then washed and overlaid with warm 0.5% (w/v) agarose or 3% Carboxymethyl cellulose (CMC) in culture medium and incubated for 24 h to 48 h. MRBΔG viruses were titered using the same protocol on T-REx™-293 cells in collagen-coated plates. G protein expression was induced by 800 μM DOX. Viral plaques were visualised by staining with 0.05% (w/v) crystal violet in 17% (v/v) methanol. Results are expressed as PFU per ml or per mg of tissue. Titration of Sindbis virus samples: Quantification of SV infectious particles was determined by plaque assay on BHK-21 cells as previously described[64].

**Synthetic lethality studies.** Cells were seeded into 24-well plates in DMEM 10% FBS. Once confluent, the cells were infected with VSVΔ51-amiR-NTC or VSVΔ51-amiR-4 by removing media, adding virus in serum-free media for 1 h, removing inoculum and adding supplemented media. Following 18-h incubation 15 μM GSK126 (Active Biochem) prepared in DMSO or equivalent volume of DMSO was added to the wells. The supernatant was collected for plaque assays, an alamarBlue® Assay was performed and cells were stained with crystal violet at the end of the experiment. A crystal violet cytotoxicity assay was then performed. See Supplementary Fig. 3a for the experimental timeline for different cell lines. For SEV transfer experiments, isolated SEVs derived from mock-infected cells or from MRBΔG-amiR-NTC- and MRBΔG-amiR-4-infected cells were transferred to a confluent monolayer of cells in 48-well plates. The treatment plan including SEV amount transferred can be found in Supplementary Fig. 4g.

**Spheroids.** To generate spheroids, poly-HEMA-coated round-bottom 96-well plates were seeded with 5E3 BxPC-3 cells/well. Plates were subjected to 15-min centrifugation set at $900 \times g$ and were incubated at 37 °C for 48 h. Spheroids were then infected with VSVΔ51-amiR-NTC or VSVΔ51-amiR-4 for an hour in serum-free media. Media was then replaced with DMEM 10% FBS. After 1, 3, 5, 7 and 9 days, GSK126 (15 μM) or the equivalent volume of DMSO was added to the spheroids. Images were taken on days 2 and 9 by transmitted light microscopy at 40X on the EVOS FL Cell Imaging System (Thermo Fisher Scientific). Acini diameter was measured using ImageJ 1.52k[65].

**Measurement of infection and cell death by flow cytometry.** HPAF-II, 786-0 and 4T1 cells were infected with VSVΔ51-amiR-4 and treated with or without GSK (15 μM) following the treatment plan outlined in Supplementary Fig. 4i before staining with polyclonal rabbit anti-VSV primary antibody (1:15000)[62] followed by anti-rabbit Alexa Fluor 647 secondary antibody (1:15000; Jackson ImmunoResearch). Stained cells were analysed on the BD LSR Fortessa (Becton Dickinson [BD], Franklin Lakes, NJ) and data were analysed using FlowJo v10 (FlowJo, LLC, Ashland, OR). FSC/SSC was used to identify cells and FSC-A/H was used to determine single cells. Gates for infected cells were set using an uninfected stained sample.

**Extracellular vesicle isolation.** EVs were collected using differential centrifugation of conditioned media. Cells were infected at MOI 5 with MRBΔG viruses in serum-free media. After 2 h, media was changed for DMEM with exosome-depleted serum (Thermo Fisher Scientific) and the cells were incubated for 24 h at 37 °C. Samples were first subjected to $2000 \times g$ and $12,000 \times g$ centrifugation steps (Thermo Scientific Sorvall ST 40 R Centrifuge and Beckman Coulter Avanti J-E, respectively) to eliminate cells, cellular debris and large EVs. Small EVs were pelleted using ultracentrifugation (Beckman Coulter Optima L-100 XP Ultra-centrifuge or Thermo Scientific Sorvall wX+ Ultra Series Centrifuge) at $120,000 \times g$ for 3 h and collected in 1× PBS.

**Nanoparticle tracking analysis.** To determine size distribution and concentration of EV preparations, nanoparticle tracking analysis (NTA) was carried out using the ZetaView® (Particle Metrix). EVs resuspended in 1× PBS were diluted from 100- to 1000-fold. To ensure consistency in concentration readouts, measurements were performed using identical settings using the ZetaView 8.03.08 [0106] software.

**Cryo-electron microscopy (Cryo-EM).** Cryo-EM grids were prepared with 3 μl of purified EV preparation onto 200 mesh grids with 2-μm holes (Quantifoil R2/2, Quantifoil Micro Tools, GmbH, Germany). Grids were glow discharged for 30 s prior to applying the sample (Cressington, UK). Grids were plunge-frozen in liquid ethane cooled by liquid nitrogen using a FEI Vitrobot IV (FEI, The Netherlands) at 90% relative humidity, and a chamber temperature of 4 °C. Micrographs were imaged using the FEI Titan Krios EM (Astbury Biostructure Laboratory, University of Leeds) at 300 kV, using a total electron dose of 60 e−/Å$^2$ and a magnification of ×75,000 and the EPU Software (Thermo Fisher). Micrographs were acquired using an energy-filtered K2 Summit direct electron detector (Gatan, USA), with a final object sampling of 1.07 Å/pixel.

**Immunofluorescence microscopy.** SEVs derived from 786-0 cells infected with MRBΔG-amiR-4 were stained with CFSE (Invitrogen) at a final concentration of 15 μM for 10 min at 37 °C. Excess dye was removed from SEVs using the Exosome Spin Columns (MW 3000; Invitrogen). SEVs were then transferred onto naive 786-0 cells grown on coverslips in a six-well plate and incubated for 2 h at 37 °C. Cells were then washed three times with 1× PBS, fixed using 1% PFA for 5 min, washed three times with 1× PBS, stained using Hoechst 33342 nucleic acid stain

(Invitrogen) for 10 min and washed three times with 1× PBS. Coverslips were then mounted on slides using Vectashield H-1000 (Vector Laboratories). Samples were imaged on the Axio imager.M1 microscope (Zeiss) with the AxioCam HRm camera (Zeiss) and the AxioVision release 4.8.2 software (Zeiss).

**RNase treatment of EVs**. To evaluate the presence of non-encapsulated RNA in the EV preparations, samples were subjected to RNase treatment. EV preparations were treated with 1 unit of RNase A for 30 min at 37 °C. EV-encapsulated RNA was then extracted using TRIzol™ reagent (Invitrogen) and subjected to RT-qPCR. A sample containing extracted RNA from 786-0 cells was used as a control following the same treatment and was resolved on a 1.2% agarose gel alongside an untreated sample to show RNase A activity. Gels were imaged on the Epi Chemi II Darkroom Imager (UVP) using the LabWorks software version 4.6.

**Transwell assay**. B16-F10 or 4T1 wild-type and Rab27a knockout cells were seeded in the upper compartment of a transwell cell culture insert with 0.4-μm pore diameter (Corning) and were infected with spread-compromised viruses (MRBΔG-amiR-4 or VSVΔ51-shPD-L1 or their respective virus controls) at MOI 3 for 2 h. The transwell cell culture inserts were then washed twice with 1× PBS and transferred to a plate containing naive (uninfected) B16-F10 or 4T1 wild-type cells in the lower compartment (Figs. 4g and 6b). The plates were incubated at 37 °C for 48 h and wild-type cells at the bottom of the transwell chamber were harvested in TRIzol™ before being subjected to RNA extraction and RT-qPCR or cells were collected for protein extraction and immunoblot analysis.

**Animal studies**. All animal studies complied with ethical regulations and were approved by the Institutional Animal Care Committee of the University of Ottawa (animal protocol # OHRI2870) and carried out in accordance with guidelines of the National Institutes of Health and the Canadian Council on Animal Care. Mice with palpable tumours were monitored daily. Mice were euthanized at the indicated experimental time point. Otherwise, animals were euthanized at the pre-established human endpoint criteria. Animals displaying signs of pain, lethargy, laboured breathing, lack of responsiveness, significant abdominal distension due to ascites build up, or when tumour volume reached 1500 mm³, were endpointed. Animals were blindly randomised to treatment groups upon tumour implantation and before therapeutic treatments. Animals that did not develop palpable tumours were excluded from the study. All animal manipulations were conducted with the operator blinded to the experimental condition and allocation group.

**Biodistribution**. To assess the biodistribution of VSVΔ51-amiR-4 and ensure no or minimal effect of the virus on normal tissues, C57BL/6 female mice, (VSVΔ51 control: 7 mice, VSVΔ51-amiR-4: 8 animals; Charles River Laboratories, Wilmington, MA) were injected IV with 1E8 pfu of VSVΔ51-amiR-4 or VSVΔ51-miR-NTC. Mice were sacrificed 48 h post injection and key organs were harvested. Liver, kidney, lungs, ovary and brain were flash-frozen, homogenised using the QIAGEN TissueLyser II and titrated by plaque assay as described above.

**Xenograft mouse models**. HPAF-II cells (1E7 cells/tumour) were injected subcutaneously (SC) in nude CD-1 female (8–12 weeks old) mice (Charles River Laboratories, Wilmington, MA). To compare viral titers of VSVΔ51-amiR-4 and VSVΔ51-amiR-NTC, 5E7 PFUs (12 animals per group) were injected intratumourally (IT) once tumours reached ~100 mm³. An equivalent volume of 1× PBS was injected in control mice. After 48 h, mice were sacrificed and tumours were collected, flash-frozen and titrated by plaque assay. To evaluate gross histopathology tumours were collected and stained using a hematoxylin and eosin (H&E) kit (Sigma-Aldrich) and imaged using the Scanscope (Aperio) and ImageScope version 11.1.2.760 software (Aperio). To compare the efficacy of VSVΔ51-amiR-4 and VSVΔ51 to control tumour growth (PBS, $n = 4$; VSVΔ51-amiR-NTC, $n = 3$; VSVΔ51-amiR-4, $n = 4$), 5E7 PFUs were injected IT on days 29–31 and 36–38. To assess the synthetic lethal effect induced by amiR-4 and GSK126, ~100 mm³ tumours were injected with VSVΔ51-amiR-4 or VSVΔ51-amiR-4 and GSK126 on days 20–22 and 27–29. GSK126 (50 mg/kg) or the equivalent volume of Captisol 20% (vehicle control; Ligand Pharmaceuticals, Inc) was injected intraperitoneally (IP) for 10 consecutive days starting on day 21. Tumour size was measured three times a week using calipers and tumour volume was calculated using a modified ellipsoidal formula; tumour volume = [(width² × length)/2], where width is the smallest dimension[66].

**Immunocompetent mouse models**. For orthotopic tumour model studies, TH04 mouse pancreatic cancer cells (1E4) were injected in the tail of the pancreas of immunocompetent C57BL/6 female (8–12 weeks old; $n = 5$ per group) mice (Charles River Laboratories, Wilmington, MA) to assess the survival advantage of VSVΔ51-amiR-4 compared to VSVΔ51-amiR-NTC. 3E8 PFUs of the viruses or equivalent volume of 1× PBS were injected IP on days 5–7 and 12–14. For the orthotopic melanoma model, B16-F10 cells (1E5 cells/animal) were injected subcutaneously (SC) in C57BL/6 female (8–12 weeks old) mice (Charles River Laboratories, Wilmington, MA). To compare the survival rates of mice (five animals per group) treated with VSVΔ51 viruses, 3E8 PFU were injected intratumourally (IP) on days 7, 9, 11, 14, 16 and 18. An equivalent volume of 1× PBS was

injected in control mice. To assess the synthetic lethal effect induced by amiR-4 and GSK126, mice were then injected with GSK126 (50 mg/kg) or the equivalent volume of Captisol 20% (vehicle control) IP for 10 consecutive days starting on day 6. To assess the combinatorial therapeutic effect induced by amiR-4 and anti-CTLA4, C57BL/6 female (8–12 weeks old) mice (five animals per group) bearing orthotopic B16-F10 tumours were treated with VSVΔ51 (3E8 PFU, IP) on days 7, 9, 11, 14, 16 and 18 and also injected with anti-CTLA4 antibodies (clone 9D9, Bio × Cell, 5 mg/kg) or the equivalent concentration of IgG isotype control (clone MPC-11 Bio × Cell) intraperitoneally (IP) during days 8–18. Tumours were grown to an average volume of ~32 mm³ before treatment. Tumour volume was calculated by measuring the length and width of each tumour using calipers, where tumour volume = [(width² × length)/2], where width is the smallest dimension[66].

For intraperitoneal tumour models, B16-F10 cells (2E5 cells/animal) were injected IP in C57BL/6 female (8–12 weeks old) mice (Charles River Laboratories, Wilmington, MA). To compare the survival rates of mice treated with VSVΔ51-amiR-4 and VSVΔ51 control, 3E8 PFUs were injected IP on days 5–7 and 12–14. The equivalent volume of 1× PBS was injected in control mice. To assess the synthetic lethal effect induced by amiR-4 and GSK126, mice were then injected with GSK126 (50 mg/kg) or the equivalent volume of Captisol 20% (vehicle control) IP for 10 consecutive days starting on day 6; ($n = 10$ for PBS + GSK126 and VSVΔ51-amiR-4 + GSK126 groups and $n = 9$ for the VSVΔ51-amiR-4 + Captisol group). For the syngeneic ovarian cancer ID8 Trp53⁻/⁻ peritoneal carcinomatosis model, 5E6 cells/animal were injected IP in female (8–12 weeks old) C57BL/6 mice (5 animals per group; Jackson Laboratories). To compare the survival rates of mice treated with VSVΔ51-amiR-4 and VSVΔ51, 3E8 PFUs were injected IP on days 7–9 and 14–16. An equivalent volume of 1× PBS was injected in control-treated mice. Mice were euthanized at a humane endpoint once severe ascites was built up in the abdomen.

**Preparation of single-cell suspension, antibody staining and flow cytometry**. B16-F10 subcutaneous tumours (PBS, $n = 3$; PBS + GSK126, $n = 4$; VSVΔ51-amiR-NTC + vehicle, $n = 4$; VSVΔ51-amiR-NTC + GSK126, $n = 4$; VSVΔ51-amiR-4+vehicle, $n = 5$; VSVΔ51-amiR-4 + GSK126, $n = 4$) were collected for immunophenotyping one day post-treatment of oncolytic virus and drug. Treatment regimen: Day 1—Captisol or GSK126 (50 mg/kg), Days 2–4—Virus (1E8 pfu) + Captisol or GSK126 (50 mg/kg), Days 5–7—Captisol or GSK126 (50 mg/kg). Tumours were dissociated using gentleMACS Dissociator and a Tumour Dissociation Kit-Mouse (Miltenyi Biotec, Auburn, CA) following the manufacturer's protocol for soft tumours. Isolated tumour cells were filtered through 70-μm cell strainers and treated with Ammonia Chloride Potassium (ACK)-lysis buffer to remove red blood cells. Single-cell suspensions were resuspended in RPMI media supplemented with 10% FBS. Then, $2 × 10^6$ cells were stained first with Fixable Viability Stain eFluor 780 (Thermo Fisher Scientific 65-0865-14, 1:1000 dilution in PBS) for 15 min at room temperature. Cells were then washed twice with fluorescence-activated cell sorting (FACS) buffer (0.5% BSA/PBS) before incubation with Purified Rat Anti-mouse CD16/CD32 (Mouse BD Fc Block™, 1:100 dilution in FACS buffer) at 4 °C to block unspecific binding of antibodies with FC receptors. Cells were stained for extracellular markers (anti-CD3-AF700, anti-CD69-BV605, anti-CD45-BV786, anti-NK1.1-APC, anti-CD49b-FITC, anti-CD122-PE and anti-CXCR3-BV421, see Supplementary Table 6 for dilution and source information) for 30 min at 4 °C. After staining, cells were washed twice with FACS buffer and fixed in 1% paraformaldehyde. Data were acquired on a Becton Dickinson (BD) flow cytometer (LSR Fortessa) and the BD FACSDIVA 6.0 software, and analyses were performed using FlowJo v10 (FlowJo, LLC, Ashland, OR).

**Multiplex immunohistochemistry and NanoString analyses**. B16-F10 subcutaneous tumours were collected for immunophenotyping five days post-treatment of oncolytic virus and drug. Treatment regimen: Day 1–Captisol or GSK126 (50 mg/kg), Days 2–4–Virus (1E8 pfu) + Captisol or GSK126 (50 mg/kg), Days 5–7–Captisol or GSK126 (50 mg/kg). Multiplexed immunohistochemistry (five tumours/animals per group) was performed on 4-μm tissue sections using Opal reagents (Akoya Biosciences, NEL811001KT) in combination with the following antibodies: CD4 (1:2000; eBioscience, 14-9766); CD8 (1:2000; eBioscience, 14-0808); Anti-TBR2-Eomes Rabbit monoclonal antibody (1:2000; Abcam, ab216870); FOXP3 (1:1000; eBioscience, 14-5773-82). Completed slides were imaged using a Vectra 3 automated imaging platform (Akoya Biosciences) and resulting images unmixed using InForm v2.4.8 software (Akoya Biosciences). Cell detection and phenotyping were performed using a custom algorithm developed in Qupath v0.2.3[67]. (QuPath: Open-source software for digital pathology image analysis. For NanoString studies, (RNAzol RT extraction reagent; Sigma) 300 ng of RNA for each tumour sample (two tumours/animals per group) and biological replicate were used as the NanoString process input with the nCounter Mouse Immunology Gene Expression Panel (XT-CSO-MIM1-12). After data collection, the background substitution and normalisation were done using solver software from NanoString. The normalised data was used as an input for differential expression analysis using Tidyverse library[68] and EdgeR package[69] in R. Heatmaply[70] as used for drawing the heatmap.

**Statistics and reproducibility**. Experimental plans, sample size and statistical analysis are provided in the main text and in the figures or figure legends. All in vitro

experiments were repeated at least twice unless otherwise stated. Mouse studies were performed twice unless otherwise stated in the figure legend. All measurements were taken from distinct biological replicate samples. Animal cohorts were randomised following tumour implantation before initiation of the treatment plan. In certain rare cases, outliers were excluded from sample analysis. When the sample size was less than three biological replicates, no statistical analyses were performed. Statistical analyses were performed using GraphPad Prism 6 and 7 (GraphPad). Quantitative data are reported as mean ± SEM or SD. as indicated in the figure legends. Statistical analysis was performed on raw data by Student's $t$ test to compare two independent conditions, one-way ANOVA to compare three conditions or more, two-way ANOVA with Tukey's or Sidak's correction to compare groups influenced by two variables, and the Kaplan–Meier method followed by log-rank test for survival analysis. The statistical significance of all $P$ values are: $^{ns}P > 0.05$, $^*P < 0.05$, $^{**}P < 0.01$, $^{***}P < 0.001$ and $^{****}P < 0.0001$. Differences between experimental groups were considered significant at $P < 0.05$. Exact $P$ values are provided in the text, figure legends or source data as indicated.

**Reporting summary**. Further information on research design is available in the Nature Research Reporting Summary linked to this article.

## Data availability

The RNA-seq data were deposited to the Gene Expression Omnibus (GEO) and is available at: [https://www.ncbi.nlm.nih.gov/geo/query/acc.cgi?acc=GSE189281], under accession no. GSE189281. cBioPortal for cancer genomics online platform [https://www.cbioportal.org/] to query the publicly available pancreatic cancer sequencing datasets (4 studies: UTSW[71]; QCMG[72]; TCGA, Firehose Legacy; Multi-Institute[73]) and generate the graph displayed in Supplementary Fig. 2h. Uncropped western blots of all data included in the main Figures and Supplementary Figures are provided in the Source Data file or in Supplementary Fig. 8, respectively. Full Targeted RNA-sequencing datasets of the unpassaged and passaged Sindbis virus library are provided in the Supplementary Data 1 file and publicly available [https://www.ncbi.nlm.nih.gov/geo/query/acc.cgi?acc=GSE197271]. NanoString datasets are included in the Supplementary Data 2 file. The remaining data are available within the Article, Supplementary Information or Source Data file. Source data are provided with this paper.

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

## Acknowledgements

This work was funded by grants from the Canadian Institutes of Health Research (#377104) and the Canadian Cancer Society Innovation (#705973) and Impact (#IMP-14 and # 706162) grants to J.C.B. and C.S.I. as well as the generous support from the Ontario Institute for Cancer Research, 71the Ottawa Regional Cancer Foundation and the Ottawa hospital foundation. J.-S.D. is supported by the CIHR New Investigator Award - Infection and Immunity (INI-147824). J.-S.D. and J.C.B. hold grants from the Terry Fox Research Institute (TFF 122868) and the Canadian Cancer Society supported by the Lotte & John Hecht Memorial Foundation (703014). We are grateful for the exceptional support from Christina Walker and the Pancreatic Cancer Purple Team to Dr. Ilkow's research. The Pancreatic Purple Team has made possible the purchase of laboratory equipment used in this study. M.J.F.C., J.P., T.R.J., Z.T., R.R., R.S., E.E.F.B. and A.S. received funding support from CanPRIME in the form of Mitacs Accelerate fellowships. M.E.W. was funded by a CIHR Frederick Banting and Charles Best Canada Graduate Scholarship and an Ontario Graduate Scholarship. M.J.F.C. is funded by the Taggart-Parkes Fellowship. T.R.J. was funded by a CIHR Frederick Banting and Charles Best Canada and Graduate Scholarship. Z.T. is funded by an NSERC CGS-D3 and an Ontario Graduate Scholarship. A.S. is supported by an NSERC Graduate Scholarship. H.E.M. was funded by a CIHR Frederick Banting and Charles Best Canada Graduate Scholarship. L.P. and R.S. were funded by CIHR postdoctoral fellowships, T.A. was funded by a CIHR Banting Fellowship., and A.P. and E.E.F.B. were funded by the Lebovic Fellowship Funding. We thank the exceptional technical support of C. Cemeus and members of the Bell, Auer, Atkins and Diallo laboratories at the Ottawa Hospital Research Institute for feedback on this project. The FEI Titan Krios microscopes of the EM facility (University of Leeds) were funded by the University of Leeds (UoL ABSL award) and Wellcome Trust (108466/Z/15/Z). The use of the ZetaView was made possible thanks to Dr. Burger at the University of Ottawa, funded by the Canada Foundation for Innovation, the Ontario Research Fund, and The Ottawa Hospital Research Institute. We thank the personnel of the Flow Cytometry and Virometry Core Facility and Histology Core Facility of the Faculty of Medicine at the University of Ottawa for their support. Graphical illustrations displayed in Figs. 1a, 4g, 6b, g and Supplementary Fig. 1a were illustrated by Christine Kennedy, Medical Illustrator [http://www.ckenneyillustration.com/]. The graphical illustration shown in Supplementary Fig. 4b was made using BioRender (License agreement number: WF23CQBING).

## Author contributions

M.-E.W., V.A.J., M.J.F.C., J.P., G.P., B.L., M.B., H.E.M., L.P., X.H., E.R., E.E.F.B., N.C., C.T.S., S.T.K., A.S., R.S., D.G.R., G.M., B.M., M.L.C., E.J.J., B.K. and C.S.I. conducted in vitro experiments. M.-E.W., M.J.F.C., J.P., T.R.J., G.P., C.T.S., J. P. and C.S.I. performed mouse experiments. Z.T., N.A., M.J.F.C. and S.T.K. performed flow cytometry experiments. M.J.F.C., J.P. and T.R.J. collected tumours for multiplex immunohistochemistry, performed by D.M. and V.A.J. M.J.F.C., J.P., T.R.J. B.A. prepared tumour samples for NanoString, using equipment from M.G. and K.J.R., and data were analysed by R.R. and T.A. A.P., T.N.Y. and P.C.B. performed RNA-seq analyses. A.C. recruited and collected biopsies from subjects with pancreatic cancer. V.A.J., J.P., M.J.F.C., G.P., A.A. and C.S.I. engineered the viruses. M.-E.W., V.A.J., M.J.F.C., J.P. and C.S.I. wrote the manuscript. B.L., R.C.A., J.-S.D., D.G., B.R.T., A.M. and J.C.B. contributed to the design of studies.

## Competing interests

We declare that Dr. John Bell has an interest in Turnstone Biologics, which develops the oncolytic Maraba MG1 virus as an OV platform. The remaining authors declare no competing interests.
