## [Peer Review File · Nature Communications]

Reviewers' Comments:

Reviewer #1:

Remarks to the Author:

I provided my feedback in my original review.

The authors have addressed all of my concerns raised in my original review.

I believe it is acceptable for publication in Nature Communications.

Reviewer #2:

Remarks to the Author:

The authors have done an admirable job of responding to the reviewer concerns and the revised manuscript is much more organized and clear to read. I have no further concerns and believe this work will be of interest to those in the field.

Reviewer #3:

Remarks to the Author:

I appreciate that the authors addressed majority of the concerns that I raised in the first round of revision. The experimental approach, execution and data presentation are satisfactory. However, while skimming through the figures, I found many typing and grammatical mistakes that need to be addressed prior to publication. Please proof-read the manuscript to make sure that figure legends correspond to right data.

1. Colony formation assay results (Figure 2C-h): Some of the wells in the figures are highly saturated with blue color. Can we rely on the quantification of those wells where the colonies are not even visible? Software-based analysis of such over-saturated wells is also not accurate.

2. Figure 2i and 2j- figure legends do not match with respective figures.

3. Supplementary Figure 7- Figure legends need proof-reading.

“(c,d) Multi-step growth curves (c) and cytotoxicity assays (d) were 1365 conducted in B16-F10 cells to evaluate growth and killing activity of VSVD51-shPDL1 and 1366 VSVD51-amiR-NTC control (MOI= 0.001).”

Response to the reviewer's comments:

Reviewer #1 (Remarks to the Author):

I provided my feedback in my original review. The authors have addressed all of my concerns raised in my original review. I believe it is acceptable for publication in Nature Communications.

We thank the reviewer for the time and effort invested into the review of our manuscript, and for the helpful comments and suggestions.

Reviewer #2 (Remarks to the Author):

The authors have done an admirable job of responding to the reviewer's concerns and the revised manuscript is much more organized and clear to read. I have no further concerns and believe this work will be of interest to those in the field.

We thank the reviewer for the time and effort invested into the review of our manuscript, and for the helpful comments and suggestions.

Reviewer #3 (Remarks to the Author):

I appreciate that the authors addressed the majority of the concerns that I raised in the first round of revision. The experimental approach, execution and data presentation are satisfactory. However, while skimming through the figures, I found many typing and grammatical mistakes that need to be addressed prior to publication. Please proof-read the manuscript to make sure that figure legends correspond to right data.

We thank the reviewer for the time and effort invested into the review of our manuscript, and for the helpful comments and suggestions. We have extensively edited and proofread the manuscript. We have also carefully verified that the figure legends correspond to the right data.

1. Colony formation assay results (Figure 2C-h): Some of the wells in the figures are highly saturated with blue color. Can we rely on the quantification of those wells where the colonies are not even visible? Software-based analysis of such over-saturated wells is also not accurate.

The data shown in Figure 2c-h is not derived from colony-forming assays, we apologize if we were not clear in the Figure legend. The data shown in these figures are crystal violet cytotoxicity assays performed as detailed in the Methods section. Crystal violet cytotoxicity assay is one of the methods commonly used to detect virus or drug cytotoxicity. We recognize that we did not make as clear as necessary the type of assay used in Figures 2c-h, we have now edited the associated figure legends to make these links more obvious.

2. Figure 2i and 2j- figure legends do not match with respective figures.

We have edited the Figure legends for both Figure 2i and 2j.

3. Supplementary Figure 7- Figure legends need proof-reading. ???(c,d) Multi-step growth curves (c) and cytotoxicity assays (d) were 1365 conducted in B16-F10 cells to evaluate growth and killing activity of VSVD51-shPDL1 and 1366 VSVD51-amiR-NTC control (MOI= 0.001).???

We have edited the Figure legend associated with Supplementary Figure 7.